# DeepScholar-Bench: A Live Benchmark and Automated Evaluation for Generative Research Synthesis

## Abstract

The ability to research and synthesize knowledge is central to human expertise and progress. A new class of AI systems—designed for generative research synthesis—aims to automate this process by retrieving information from the live web and producing long-form, cited reports. Yet, evaluating such systems remains an open challenge: existing question-answering benchmarks focus on short, factual answers, while expert-curated datasets risk staleness and data contamination. Neither captures the complexity and evolving nature of real research synthesis tasks. We introduce DeepScholar-bench, a live benchmark and automated evaluation framework for generative research synthesis. DeepScholar-bench draws queries and human-written exemplars from recent, high-quality ArXiv papers and evaluates a real synthesis task: generating a related work section by retrieving, synthesizing, and citing prior work. Our automated framework holistically measures performance across three key dimensions—knowledge synthesis, retrieval quality, and verifiability. To further future work, we also contribute DeepScholar-ref, a simple, open-source reference pipeline, which is implemented on the LOTUS framework and provides a strong baseline. Using DeepScholar-bench, we systematically evaluate prior open-source systems, search agents with strong models, OpenAI's DeepResearch, and DeepScholar-ref. We find DeepScholar-bench is far from saturated: no system surpasses a geometric mean of $31\%$ across all metrics. These results highlight both the difficulty and importance of DeepScholar-bench as a foundation for advancing AI systems capable of generative research synthesis.

## 1 Introduction

A core foundation of human knowledge and innovation is the ability of human experts to *research and synthesize* known facts and new findings, enabling others to comprehend, verify and build upon prior work. Recently, systems for *generative research synthesis* have emerged, promising to automate tasks that produce long-form outputs (e.g., multi-page reports), which traditionally demand hours of literature searching, reading and writing by human experts. These offerings include commercial ones—from OpenAI (OpenAI, 2025a), Gemini (Gemini, 2025a), Anthropic (Anthropic, 2025a), Grok (xAI, 2025), and Perplexity (Perplexity, 2025)—as well as open-source methods, such as STORM (Shao et al., 2024), DeepResearcher (Zheng et al., 2025), and OpenScholar (Asai et al., 2024). Existing systems demonstrate promising performance on factuality and question-answering benchmarks (Wei et al., 2024; Krishna et al., 2025; Mialon et al., 2023; Wei et al., 2025), pushing the frontier of AI capabilities.

Yet, as this new class of systems emerges, a key question remains: *how should we benchmark and evaluate generative research synthesis?* The progress of these systems requires benchmarks that carefully evaluate their critical capabilities—specifically, three core functions: (1) *retrieval*, typically from a large, complex, and constantly-evolving corpus, such as the live web, to collect key information (2) *knowledge synthesis*, to generate coherent, long-form answers that surface key facts, integrating general knowledge and findings from many retrieved sources, and (3) *verifiability*, providing citations that allow readers to trace each stated claim in the synthesized answer to a reputable source from the retrieved set. The ideal benchmark must holistically evaluate across all three of these dimensions, while providing a realistic and challenging research synthesis task.

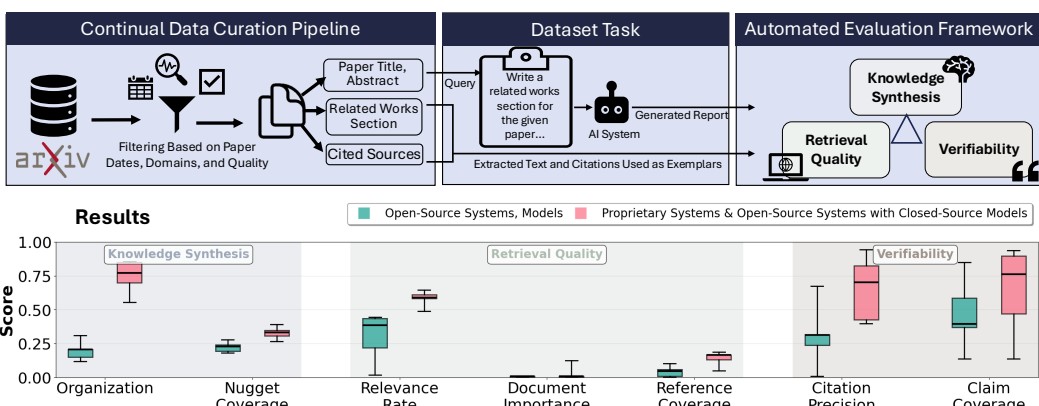

Figure 1: DeepScholarBench Overview. We propose a live, continually-updating benchmark for generative research synthesis, for which we plan to release monthly datasets and leaderboard results. We use an automated data pipeline (top left) to curate datasets from recent, high-quality ArXiv papers. Our dataset task is to generate a related works section given information about a paper (top middle). The DeepScholar-bench evaluation framework (top right) uses a holistic set of automated metrics to assess performance of system reports on three key dimensions: knowledge synthesis, retrieval quality and verifiability. We systematically evaluate 14 existing baselines (bottom) and show the performance range of them on each metric. In pink, we show the performance range of open-source systems, including DeepScholar, STORM, OpenScholar, a Search Agent and our DeepScholar-ref pipeline, each with the open-source Llama-4-Scout-17B-16E-Instruct model. In green, we show the performance range of proprietary systems and open-source systems using closed-source models, including OpenAI's o3 DeepResearch, as well as Search Agents and DeepScholar-ref run with o3, Claude-opus-4, Gemini-2.5-pro and GPT4.1. Overall, no system surpasses a geometric mean of 31% across all metrics, reflecting significant opportunity for future work. Full evaluation results appear in Section 6.

Unfortunately, existing benchmarks fall short of these goals. Many prior works evaluate generative research synthesis systems using existing question answering benchmarks, which do not reflect realistic research synthesis tasks and instead focus on questions with short-form, easily-verifiable answers, making them severely limited for this setting (Wei et al., 2025; 2024; Mialon et al., 2023; Krishna et al., 2025; Wu et al., 2025; Wadden et al., 2020; Jin et al., 2019; Yang et al., 2018; Joshi et al., 2017; Kwiatkowski et al., 2019; Ho et al., 2020; Trivedi et al., 2022; Lee et al., 2023). These question-answering benchmarks do not capture the complexity of long-form answers synthesized from many sources, a key component of research synthesis. To address this limitation, several recent works instead leverage expert-curated datasets with open-ended research questions and exemplar answers (Asai et al., 2024; Zheng et al., 2024; you.com, 2025; Xu et al., 2025; Du et al., 2025; Su et al., 2025; Java et al., 2025). Unfortunately, these benchmarks quickly become stale and outdated as new information emerges. Furthermore, these datasets risk data contamination as new models are trained on snapshots of the web, including public datasets. The prohibitive expense of curating, maintaining, and updating expert-curated benchmarks further limits their utility towards realistic, scalable evaluation.

In this work, we introduce DeepScholar-bench, a live benchmark and holistic, automated evaluation framework designed to evaluate generative research synthesis. DeepScholar-bench draws queries from recent, high-quality ArXiv papers and focuses on a real research synthesis task: generating the related work sections of a paper by retrieving, synthesizing, and citing prior research. We plan to provide a *live* benchmark, releasing updated research queries every month, and practitioners can also run our automated date pipeline to create their own dataset instantiations. Further, we develop an automated evaluation framework that leverages human-written related works extracted from each ArXiv paper and holistically assesses performance across three key dimensions— *knowledge synthesis*, *retrieval quality*, and *verifiability*—using metrics that show strong agreement with human judgments. To promote future work, we also develop DeepScholar-ref, a simple open-source reference pipeline for generative research synthesis implemented on the LOTUS framework (lotus, 2025; Patel et al., 2025).

Using the DeepScholar-bench framework, we systematically evaluate the performance of existing systems, including open-source research synthesis systems, search agents with strong proprietary models, OpenAI DeepResearch, and DeepScholar-ref. We find that all of these existing methods exhibit significant opportunity for improvement, with no system surpassing a geometric mean of 31% across all metrics. Furthermore, on several key metrics, including Nugget Coverage, Reference Coverage and Document Importance, each evaluated method's performance remains well below 40%, reflecting the inherent difficulty of the DeepScholar-bench task, which requires systems to navigate the live web, reasoning about the relevance and importance of documents as well

Table 1: Summary of Evaluation Metrics.

| Metric | Description |
|---|---|
| *Knowledge Synthesis* | |
| Organization & Coherency | assesses organization and coherence of system answer |
| Nugget Coverage | assesses the answer's coverage of essential facts |
| *Retrieval Quality* | |
| Relevance Rate | measures avg. relevance among all referenced sources |
| Document Importance | measures how notable referenced sources are, using citation counts |
| Reference Coverage | assesses the referenced set's coverage of key, important references |
| *Verifiability* | |
| Citation Precision | measures percent of cited sources that support their accompanying claim |
| Claim Coverage | measures percent of claims that are fully supported by cited sources |

as surfacing key facts into a cohesive final answer. Notably, OpenAI's DeepResearch offers strong performance relative to other baselines, outperforming many prior methods on knowledge synthesis and retrieval quality, with scores of $39.2\%$ on Nugget Coverage, $18.7\%$ on Reference Coverage and $12.4\%$ on Document Importance; however, it struggles to provide strong verifiability relative to many other methods. We also find that DeepScholar-ref reference pipeline represents a strong open-source baseline offering competitive performance on most metrics and up to $6.3\times$ higher verifiability compared to OpenAI's DeepResearch. Nevertheless, DeepScholar-bench remains far from saturated, representing exciting opportunities for further work. We hope that our benchmark framework and reference pipeline support the progress of new systems, and we believe that resolving DeepScholar-bench represents a critical milestone towards more capable AI systems.

Overall, our main contributions are the following:

- We propose DeepScholar-bench, a live benchmark dataset with real research synthesis tasks and an automated, holistic evaluation.

- We develop DeepScholar-ref, a simple open-source reference pipeline for generative research synthesis that attains competitive performance with open-source systems, search agents, and OpenAI's DeepResearch across many metrics using the same models.

- We perform a systematic evaluation of existing baselines on DeepScholar-bench, finding significant opportunities for improvement, with no system surpassing a geometric mean of $31\%$ across all metrics.

## 2 RELATED WORK

***Long-form Synthesis Benchmarks.*** While our work proposes a continually-updated, live benchmark using an automated data pipeline, several prior works instead provide expert-curated datasets for long-form research synthesis tasks, including ScholarQABench (Asai et al., 2024), OpenResearcher (Zheng et al., 2024), DeepConsult (you.com, 2025), ResearcherBench (Xu et al., 2025), DeepResearch Bench (Du et al., 2025), Deep Research Bench (FutureSearch et al., 2025), SurGE (Su et al., 2025), and LiveDRBench (Java et al., 2025). Unfortunately, these expert-curated benchmarks, are expensive to construct and update, can quickly become outdated, as new information becomes available, and risk data contamination, as new models are trained on publicly available data.

Alternatively, several recent benchmarks, including AcademicEval (Zhang et al., 2024b), LongBench-Cite (Zhang et al., 2024a) and SciIG (Garg et al., 2025), evaluate long-form generation tasks *that do not require search over the live web*, which is a key component of generative research synthesis and our benchmark. Other benchmarks focus on other long-form generation tasks, such as Wikipedia-like article generation (Shao et al., 2024), which differs substantially from our focus on complex research synthesis tasks. Crucially, unlike each of these prior works, our work proposes a live, continually-updated benchmark for evaluating generative research synthesis.

***Factuality and Question Answering Benchmarks.*** While this work proposes a framework for studying complex, long-form research synthesis tasks, which lack an absolute notion of correctness and admit many possible reasonable answers, several recent works focus their evaluation on question answering (QA) and factuality benchmarks with short-form, easily verifiable answers. These prior benchmarks include SimpleQA (Wei et al., 2024), FRAMES (Krishna et al., 2025), GAIA (Mialon

et al., 2023), BrowserComp (Wei et al., 2025), BrowserComp-Plus (Chen et al., 2025) WebWalk-erQA (Wu et al., 2025), DeepResearch Arena (Wan et al., 2025) and others traditionally used to evaluate retrieval-augmented generation (Wadden et al., 2020; Jin et al., 2019; Yang et al., 2018; Joshi et al., 2017; Kwiatkowski et al., 2019; Ho et al., 2020; Trivedi et al., 2022; Lee et al., 2023). Additionally, several benchmark develop automated dataset curation pipelines for live benchmark; however, their task focuses on short-form question-answering, as opposed to long-form report generation (Ouyang et al., 2025; Meem et al., 2024; Jiang et al., 2025).

## 3 THE DEEPSCHOLAR DATASET

We study the task of generating a related works section of an academic paper, a fundamental research synthesis task, for which we leverage human-written exemplars from our automated data pipeline to ground our evaluation. We source our dataset queries by scraping ArXiv papers (arxiv, 2025) accepted at academic conferences, and we formalize our dataset task as follows: given a paper's description $d$, the goal is to retrieve a set of relevant sources $S$ and generate a related works sections $W$ by synthesizing and citing the retrieved documents. We briefly overview our automated data collection framework (Section 3.1) and describe the dataset instantiation (Section 3.2) used in our evaluation (Section 6). Appendix 8.1 provides additional details.

### 3.1 AUTOMATED DATA COLLECTION FRAMEWORK

Our automated data collection framework aims to achieve the following design goals:

1. Inclusion of *diverse* paper topics across a wide variety of research domains.

2. Focus on *recent* research papers, both to provide realistic, timely benchmark queries and to prevent data contamination when benchmarking models trained on snapshots of the web.

3. Control for *quality* of the scraped ArXiv papers and extracted data, focusing on peer-reviewed manuscripts that are accepted at academic conferences.

Our data pipeline scrapes papers, filters them, and extracts content to construct datasets that include metadata about each ArXiv paper (e.g., the title, abstract, and link), the papers' related work section, and a citation list of references found in the papers' related work sections. Our scraper begins by loading papers from a configured set of ArXiv domains (e.g., cs.ML) and configured publication-date range. To avoid possible data contamination arising from multiple ArXiv versions, we keep only v1 ArXiv papers. To control for paper quality, our pipeline then optionally filters papers to keep only those listed as "accepted" or "published" at a conference based on the ArXiv metadata. We also exclude papers that do not have an explicit "Related Work" section and .bib file, containing well-formatted bibliography entries. Our pipeline then processes each paper to extract the Related Works section from both the LaTex files and PDF file, if available. We clean the extracted LaTex section to remove any labels and comments. We also extract all citations found in the related work section and we use the ArXiv and OpenAlex APIs (OpenAlex, 2025) to recover more detailed information, such as abstracts, authors, and links for both ArXiv and non-ArXiv references.

### 3.2 DATASET DESCRIPTION AND STATISTICS

The dataset instantiation, DeepScholar-June-2025 used in our evaluation (Section 6) configures a publication date range between April and June 2025, following the April 5th, 2025 release date of Llama-4 models (noa, 2025), the main open-source model we benchmark. This instantiation scrapes papers from a diverse set of 18 distinct ArXiv domains—including, cs.IR, cs.CV, cs.AI, cs.CL, cs.LG, cs.DC, cs.DB, cs.AR, cs.SD, cs.CR, cs.ET, cs.GR, cs.PL, cs.SY, cs.OS, cs.PF, cs.SE, cs.MM—and selects papers marked as accepted at a conference. We additionally exclude papers with related works sections longer than 1,000 words to control for cost. Our final dataset includes 63 ArXiv papers, each providing a single query and an extracted expert-written exemplar for our evaluation. We make our scripts available to allow others to configure different datasets, and we plan to release monthly datasets of recent queries. Our experiments leverage the abstract of each paper as the paper's description $d$, provided to each baseline system as context within the query. We analyze the human-written exemplars from our dataset, and we find that, on average, each related work

section contains 23 unique references, and we find over 63% of all cited references on ArXiv. We provide additional experiments in Appendix 8.3.2 on a more recent, expanded dataset DeepScholar-Nov-2025, which contains 200 queries from over 75 distinct arXiv subject areas, spanning Computer Science, Physics, Quantitative Biology, Economics, and Quantitative Finance. Our results on this dataset confirm the generalization of our benchmark and main experimental conclusions.

# 4 THE DEEPSCHOLAR EVALUATION FRAMEWORK

Evaluating research synthesis is inherently challenging due to the task's complexity, and the lack of a simple, "ground truth" notion of correctness, permitting many possible answers. To address these challenges, we propose a holistic, automated evaluation that assesses 7 fine-grained metrics across three key dimensions that reflect the core capabilities of research synthesis: *knowledge synthesis* (Section 4.1), *retrieval quality* (Section 4.2), and *verifiability* (Section 4.3). We overview our evaluation framework in this section, and provide further details and analysis of our metrics in Appendix 8.4, as well as manual validation of our LLM-based metrics in Appendix 8.3.

## 4.1 KNOWLEDGE SYNTHESIS

Knowledge synthesis reflects the ability of a system to generate an effective final report which surfaces key facts and information into a coherent writeup. We evaluate both the overall *Organization and Coherency* of each system report, as well as its factual content according to its *Nugget Coverage*.

***Organization and Coherency.*** We use an LLM-as-a-judge to assess organization and coherence and perform pairwise comparisons between the system generated report and the human-written exemplar from the dataset. We permute each evaluated pair of reports to avoid position bias (Li et al., 2025), and we report the win-rate of each baseline. This model-based evaluation provides scalability while also serving as a strong surrogate for human preferences (Rahmani et al., 2024b; Li et al., 2024; 2025; Arabzadeh and Clarke, 2025), which we validate in our experiments in Appendix 8.3.

***Nugget Coverage.*** To assess the quality of the information content of a generated report, we use a nugget-based evaluation. An *information nugget* is an essential fact or component relevant for an answer (Pradeep et al., 2025; Upadhyay et al., 2024b;a; Faggioli et al., 2023; Rahmani et al., 2024a). For our task, we generate nuggets from the human-written exemplar related-work section for each query, and for each generated report, we compute the nugget coverage score, the fraction of nuggets present in the answer, following the automated, LLM-based methodology of Pradeep et al. (2025).

## 4.2 RETRIEVAL QUALITY

A key component of generative research synthesis is retrieval over the live web, which differs substantially from traditional information retrieval evaluations (Thakur et al., 2021; Nanni et al., 2017). This setting lacks a closed corpus with gold labels — expert-written exemplars provide *one* reasonable reference set, but there may be many possible alternative sets that are likewise high-quality. To address these challenges, we evaluate three metrics of each generated report's retrieved reference set: the relevance rate, reference coverage of key sources, and document importance.

***Relevance Rate.*** We asses the relevance of each retrieved document, following the Cranfield model (Voorhees, 2009), which is standard in IR evaluations and considers relevance of individual documents given a query, independent of other documents. We use an LLM-as-a-judge approach to assign graded relevance scores to each retrieved source, following recent works (Upadhyay et al., 2024b; Faggioli et al., 2023; Rahmani et al., 2024b; Thomas et al., 2024; Asai et al., 2024). Specifically, the LLM-judge assigns a relevance score, $Rel(s)$, from $0-2$ for each source, $s$, in the retrieved set $S$, and we compute the relevance rate of $S$ as:

$$RR(S) = \frac{1}{2|S|} \sum_{s \in S} Rel(s).$$

***Reference Coverage.*** We introduce a metric to measure the *reference coverage* of each report's retrieved set. A key challenge in measuring this value is in defining a core set of "important" sources that a good report should reference. To build this set, we take all references from the high-quality, human-written exemplar and label each as either "important" or "not-important", considering a "not-important" reference as one that could be omitted or substituted by a different source. For each

Figure 2: Overview of DeepScholar-ref. The system iteratively writes queries and performs web search, before passing the search results through series of semantic operators using the LOTUS system for LLM-based data-processing, including filtering step to discard irrelevant sources, a top-k ranking step to find most relevant sources, and an aggregation step to generate the final report from all remaining sources.

generated report, we then report its reference coverage by dividing the number of retrieved important references by total number of important references. We follow the below formula, where $E$ is the set of "important" references from the human-written exemplar:

$$RC(S, E) = \frac{1}{|E|} \sum_{s \in S} I[s \in E].$$

***Document Importance.*** While the above retrieval metrics assess topical relevance and coverage of key references, an ideal research synthesis system must also retrieve many *notable and important* sources. Exemplar human-written reports typically contain ample references of primary-sources and highly-cited academic publications. We compute the *document importance* of a retrieved set by considering the number of citations of each of its sources. Specifically, we consider the median number of citations over sources in $S$, compared to this the median number of citations over sources in the reference set, $S^*$, from the human-written exemplar, and set an upper-bound of 1, given by:

$$DI(S, S^*) = min\left( \frac{\text{median}\{\text{num-cites}(s)|s \in S\}}{\text{median}\{\text{num-cites}(s^*)|s^* \in S^*\}}, 1 \right),$$

where $\text{num-cites}(s)$ is the number of citations for source $s$.

## 4.3 VERIFIABILITY

To evaluate the verifiability generated reports using the citation precision and claim coverage with LLM-based entailment evaluations, following prior work (Gao et al., 2023; Liu et al., 2023).

***Citation Precision.*** We measure sentence-level precision, where a citation is considered precise if the referenced source supports at least one claim made in the accompanying sentence. For a full report, we compute citation precision by averaging the precision of each citation.

***Claim Coverage.*** Claim coverage of a report is computed by assigning a sentence-level score—of one if the sentence's cited sources support all claims made in the sentence—and averaging all sentence-level scores in the report. We make two adaptations to the original definition of prior work (Gao et al., 2023; Worledge et al., 2024; Liu et al., 2023) to tailor our metric to our long-form synthesis task. First, we relax the original claim coverage definition to consider a sliding window of sentences with supporting references Specifically, we assign a sentence-level coverage score of 1 if the sentence is fully supported by the sources cited either within the sentence or in a window of $w$ preceding or following sentences. Additionally, since our task query provides context describing a paper, we consider this context as an implicitly cited reference for each sentence.

## 5 DEEPSCHOLAR-REF

We introduce DeepScholar-ref, an open-source reference pipeline designed for generative research synthesis. As Figure 2 shows, DeepScholar-ref takes a user's query and iteratively generates web-search queries, summarizing search results in each round before generating new queries. The system then post-processes the search results leveraging a series of semantic operators (Patel et al., 2025) implemented on the LOTUS API (lotus, 2025). This includes a semantic filtering step, which leverages an LLM to filter out irrelevant source documents, followed by a semantic top-k which performs an LLM-based ranking over the documents based on their relevance to the user query. Finally, we perform a semantic aggregation over the final source documents to generate the final report. We provide further details of each step of our reference pipeline in Appendix Section 8.6.

Table 2: Main Results. The **best baseline** is shown in bold and the second-best baseline is underlined. * indicates that the best baseline is statistically significantly better than the second-best baseline under a paired two-tailed t-test with $p < 0.05$.

| | Knowledge Synthesis | | Retrieval Quality | | | | Verifiability | | Geo. Mean |
|---|---|---|---|---|---|---|---|---|---|
| | Org. | Nug. Cov. | Rel. Rate | Ref Cov. | Doc Imp. | Cite-P | Claim Cov ($w = 1$) | | |
| *Human-written Exemplars* | | | | | | | | | |
| Human-written Exemplars | .500 | 1.000 | .585 | 1.000 | 1.000 | .900[1] | .850[1] | | .782[1] |
| *Open Source Research Systems* | | | | | | | | | |
| DeepResearcher (Llama-4) | .206 | .230 | .385 | .047 | .008 | .312 | .396 | | .137 |
| STORM (Llama-4) | .119 | .183 | .218 | .003 | .006 | .238 | .586 | | .073 |
| OpenScholar (Llama-4) | .309 | .278 | .017 | .008 | .013 | .010 | .138 | | .042 |
| *Search Agents* | | | | | | | | | |
| Search Agent (Llama-4) | .151 | .193 | .445 | .060 | .009 | .316 | .368 | | .135 |
| Search Agent (GPT-4.1) | .556 | .265 | .490 | .050 | .009 | .498 | .470 | | .186 |
| Search Agent (o3) | .849 | .348 | .610 | .165 | .026 | .425 | .495 | | .287 |
| Search Agent (Claude) | .698 | .307 | .583 | .131 | .008 | .701 | .760 | | .256 |
| Search Agent (Gemini) | .706 | .277 | .583 | .061 | .010 | .415 | .398 | | .196 |
| *Commercial Systems* | | | | | | | | | |
| OpenAI DeepResearch | **.857** | **.392**\* | .629 | **.187**\* | **.124**\* | .399 | .138 | | **.309**\* |
| *DeepScholar Reference Pipeline* | | | | | | | | | |
| DeepScholar-ref (Llama-4) | .206 | .241 | .436 | .103 | .008 | .674 | .851 | | .195 |
| DeepScholar-ref (GPT-4.1) | .809 | .348 | .590 | .166 | .008 | .788 | .899 | | .285 |
| DeepScholar-ref (GPT-4.1, o3) | **.857** | .384 | **.645** | .167 | .007 | .824 | .760 | | .285 |
| DeepScholar-ref (GPT-4.1, Claude) | .698 | .307 | .610 | .152 | .009 | **.944**\* | .895 | | .286 |
| DeepScholar-ref (GPT-4.1, Gemini) | .770 | .331 | .590 | .181 | .006 | .904 | **.937**\* | | .282 |

[1] The automated verifiability metrics in our evaluation under-estimate the actual verifiability of human writing, thus, we provide an estimate using manual validation over a small sample, and we disclude them from the geometric mean for the human-written exemplars. This is because Citation Precision and Claim Coverage require us to assess entailment relations between claims and cited reference. For each LLM-based system, we are able to track the precise snippet and context from cited sources, which are directly fed as context to the LLM. On the other hand, for the human-written exemplars, we lack gold labels pointing to the precise snippet of text that each reference refers to. Our measurements for the human-written exemplars instead rely on the title and abstract of each cited source as a proxy.

## 6 EXPERIMENTAL RESULTS

In this section, we evaluate recent state-of-the-art generative research systems as well as DeepScholar-ref on DeepScholar-bench. Overall, we find the following:

- Existing baselines for generative research synthesis, including strong open-source LLM systems, search agents, and commercial systems, demonstrate significant room for improvement across all three key dimensions: knowledge synthesis, retrieval quality and verifiability. Specifically, no system surpasses a geometric mean of $31\%$ across all metrics.

- DeepScholar-ref provides a strong baseline, consistently improving upon the performance of prior open-source systems and search agents, as well as achieving competitive performance and up to $6.3\times$ higher verifiability compared to OpenAI's DeepResearch.

***Experimental Setup.*** We benchmark open-source research systems, including DeepResearcher (Zheng et al., 2025), STORM (Shao et al., 2024) and OpenScholar (Asai et al., 2024), search agents, with Llama-4-Scout-17B-16E-Instruct (Meta, 2025), GPT-4.1-2025-04-14 (OpenAI, 2025b), o3-2025-04-16 (OpenAI, 2025b), Claude-opus-4-20250514 (Anthropic, 2025b), and Gemini-2.5-pro (Gemini, 2025b) models, OpenAI's o3-deep-research (OpenAI, 2025b), and DeepScholar-ref. For each benchmarked method, we control the retrieval corpus by allowing each system to access the Web only through the ArXiv API (arxiv, 2025). We additionally avoid possible information leakage during search by filtering out any search results that were published after the query paper's publication date. We provide further details of our setup in Appendix 8.2.

### 6.1 MAIN RESULTS

Table 2 provides of each method's performance on DeepScholar-June-2025. For each baseline, we report metrics averaged over all queries, and the geometric mean of all metrics. We also provide additional results in Appendix 8.3—including metadata statistics of the generated reports (Table 6), statistics related to our evaluation metrics (Table 7), and manual validation of our LLM-based metrics (Table 10)—and example reports in Appendix 8.7. We discuss key findings in detail below.

### 6.1.1 GENERATIVE RESEARCH SYNTHESIS SYSTEMS DEMONSTRATE LARGE ROOM FOR IMPROVEMENT.

From Table 2, we see that no system surpasses a geometric mean of .31 across all metrics, with OpenAI DeepResearch obtaining the highest geometric mean. Moreover, on several key metrics, including Nugget Coverage, Reference Coverage and Document Importance, each baseline's performance remains below .40. This reflects the inherent difficulty of the generative research task provided by DeepScholar-bench, particularly the need for systems to navigate the live web, reasoning about coverage and importance of documents, before surfacing key information in long-form report.

We now analyze each evaluated dimension, comparing performance of the open-source research systems, search agents and commercial systems to the human-written exemplars. On Knowledge Synthesis, we see that OpenAI DeepResearch offers the best performance compared to all other prior methods on both Organization, with a score of .857, and Nugget Coverage, with a score of .392. OpenAI DeepResearch, as well as the o3, Claude and Gemini search agents achieve relatively high Organization scores compared to human-written exemplars. However, on Nugget Coverage all prior methods scores below .40. This demonstrates that while existing systems, especially those using state-of-the-art models, can generate well-organized and coherent summaries, they still struggle to surface key facts to answer the research query, a crucial capability for research synthesis tasks.

Turning our attention to the Retrieval Quality performance of prior methods, we once again find significant room for improvement. Once again, OpenAI DeepResearch offers the strongest performance among the other benchmarked prior methods on Relevance Rate, Reference Coverage and Document Importance, but still far from saturates performance. While it's Relevance Rate shows strong performance, exceeding that of the human exemplars with a score of .629, it's Reference Coverage and Document Importance scores remain exceedingly low: .187 and .124 respectively. This demonstrates that while state-of-the-art generative research synthesis systems excel at retrieving *relevant* sources, they still struggle to find *comprehensive sets of notable sources*, falling short compared to the ability of human experts.

Lastly, we analyze the Verifiability performance of prior methods. We see that OpenAI DeepResearch is outperformed on both Citation Precision and Claim Coverage by the search agents with GPT4.1, o3, Claude and Gemini models. The Claude search agent offers the highest Citation Precision, a score of .701 and Claim Coverage, a score of .760. Meanwhile, OpenAI's DeepResearch as well as the all other prior methods are unable to achieve a Citation Precision score beyond .50 and a Claim Coverage score beyond .60. We also note that the human-written exemplars appear to exhibit rather low Citation Precision and Claim Coverage scores, however these scores are under-estimate the actual verifiability of human writing[1]. Overall, we see that prior LLM-based systems exhibit significant headroom for improvement.

### 6.1.2 DEEPSCHOLAR-REF PROVIDES A STRONG BASELINE FOR GENERATIVE RESEARCH SYNTHESIS.

We compare the performance of DeepScholar-ref to OpenAI DeepResearch, search agents and open-source systems, finding that DeepScholar-ref provides a strong baseline with competitive performance against the other baselines across most metrics using the same or cheaper models. In comparison to OpenAI DeepResearch, DeepScholar-ref (GPT-4.1, o3) achieves a similar or higher Organization, Nugget Coverage, Relevance Rate, Reference Coverage, Citation Precision and Claim Coverage score. Notably, DeepScholar-ref achieves up to $6.3\times$ higher Verifiability scores but its Document Importance remain relatively low compared to OpenAI's DeepResearch. In Appendix Table 6, we provide additional cost analysis, finding that DeepScholar-ref (GPT-4.1, o3) offers an efficient reference pipeline that is $4.3\times$ cheaper and $2.28\times$ faster than OpenAI DeepResearch.

Next, we compare the performance of DeepScholar-ref and search agents, finding DeepScholar-ref offers competitive and often stronger performance—specifically, averaged across the 5 baselines, using the same primary model, DeepScholar-ref increases Organization by $1.18\times$, Nugget Coverage by $1.17\times$, the Relevance Rate by $1.06\times$, Reference Coverage by $2.03\times$, Citation Precision by $1.83\times$ and Claim Coverage by $1.86\times$. Lastly, we compare DeepScholar-ref (Llama-4) to the open-source research systems, all run with the Llama-4. We see that the prior open-source research systems exhibit trade-offs among the Knowledge Synthesis, Retrieval Quality and Verifiability dimensions.

Table 3: Ablation Study Comparing The Effect of Different Retrieval APIs. The **best baseline** is shown in bold and the second-best baseline is underlined. $^*$ indicates that the best baseline is statistically significantly better than the second-best baseline under a paired two-tailed t-test with $p < 0.05$.

| | Knowledge Synthesis | | Retrieval Quality | | | Verifiability | | Geo. Mean |
| | Org | Nug. Cov. | Rel. Rate | Ref Cov. | Doc Imp. | Cite-P | Claim Cov ($w = 1$) | |
|---|---|---|---|---|---|---|---|---|
| *DeepScholar-ref (GPT-4.1, Claude)* | | | | | | | | |
| arxiv.org Retrieval | .698 | .307 | .610 | .152 | .009 | .944 | .895 | .286 |
| parallel.ai Retrieval | .865 | .444 | .675 | .160 | .017 | .846 | .781 | .334 |
| taviliy.com Retrieval | **.929**$^*$ | .327 | .550 | .070 | .015 | .711 | .578 | .258 |
| Oracle Retrieval (arxiv.org) | .782 | .487 | **.686** | **1.000**$^*$ | **1.000** | **.955** | **.899**$^*$ | **.808** |
| Oracle Retrieval (All) | .778 | **.528**$^*$ | .680 | **1.000**$^*$ | .822 | .941 | .828 | .782 |
| *DeepScholar-ref (Llama-4)* | | | | | | | | |
| arxiv.org Retrieval | .206 | .241 | .436 | .103 | .008 | .674 | .851 | .195 |
| parallel.ai Retrieval | **.246** | .265 | .559 | .114 | .015 | .223 | .543 | .186 |
| taviliy.com Retrieval | .111 | .229 | .532 | .030 | .016 | .442 | .676 | .153 |
| Oracle Retrieval (arxiv.org) | .202 | .316 | .681 | **1.000**$^*$ | **1.000** | .658 | .868 | .590 |
| Oracle Retrieval (All) | .198 | **.350** | **.693** | **1.000**$^*$ | .822 | **.796**$^*$ | **.890** | **.600** |

Compared to the best-performing prior open-source methods for each metric, DeepScholar-ref offers competitive Knowledge Synthesis performance, $1.09\times$ higher Relevance Rates, $2.18\times$ higher Reference Coverage, $2.08\times$ higher Citation Precision and $1.41\times$ higher Claim Coverage.

Overall, the strong *relative* performance of DeepScholar-ref likely reflects the efficiency of the data-processing semantic operators (Patel et al., 2025) that DeepScholar-ref uses to perform LLM-based filtering, ranking and summarization of sources to generate its report. Notably, DeepScholar-ref still demonstrates significant room for improvement and far from saturates DeepScholar-Bench, especially on key Knowledge Synthesis and Retrieval Quality metrics.

## 6.2 UNDERSTANDING OPPORTUNITIES FOR IMPROVEMENT.

To analyze performance and opportunities for improvement, we conduct an ablation study, testing different retrievers, including two oracle settings. Table 3 shows the performance of DeepScholar-ref (GPT-4.1, Claude) and DeepScholar-ref (Llama-4), each with 3 different retrieval APIs: arxiv.org, the default in our main results, parallel.ai and tavily.com. In addition, our two oracle retrievers include the Oracle Retrieval (arxiv.org) setting and the Oracle Retrieval (All) setting, which provide the system with *ArXiv* references and *all* references, respectively, from the set of important references cited in exemplars, following our methodology for evaluating Reference Coverage.

Overall, the results demonstrate that key opportunities for improvement lie in both retrieval and knowledge synthesis capabilities. First, we see that DeepScholar-ref (GPT-4.1, Claude) with either oracle retriever nearly saturates performance on Retrieval Quality and Verifiability metrics, whereas the same method, using the arxiv.org, parallel.ai or taviliy.com retrievers, score much lower. Specifically, significant performance gaps exist for Reference Coverage and Document Importance, demonstrating the system struggles to navigate the live web and recover a diverse set of key, notable sources. Additionally, we also see that oracle retrievers improve the Nugget Coverage of either DeepScholar-ref methods by up to $1.62\times$ compared to the arxiv.org, parallel.ai or tavily.ai retrievers. Yet, the oracle settings still far from saturate Nugget Coverage, highlighting the AI system still struggle to effectively surface important facts and insights, even with high-quality sources.

## 6.3 HUMAN EVALUATION

To assess whether LLM-based metrics reflect human expert judgments, we conduct a human evaluation with 11 annotators, all Computer Science PhD students from four research universities across North America. In total, we collect over 300 human annotations in order to validate the LLM-based judges introduced by our automated evaluation for assessing knowledge synthesis and retrieval quality. We provide further details of our setup in Appendix 8.3.4.

**Agreement analysis.** Table 4 shows the results of human evaluation study as confusion matrices taken between the majority vote of human annotators and the LLM-judge. Overall, the results show the robustness of our LLM-judges based on their strong alignment with the expert human annotators. Specifically, we observe a 71.43% agreement score for pairwise comparisons judging Organization, a 83.33% agreement score for nugget labeling to compute Nugget Coverage, and a 65.9%

Table 4: Confusion matrices comparing human and LLM judgments on organization, Nugget importance judgments, and Reference-importance judgments, where in each table rows and columns represent human and LLM judgments.

| Organization | | | |
|---|---|---|---|
| **Human / LLM** | Reference | Generated | Tie |
| Reference | 14.29% | 0% | 14.29% |
| Generated | 0% | 57.14% | 14.29% |
| Tie | 0% | 0% | 0% |

| Nugget Importance | | |
|---|---|---|
| **Human / LLM** | Vital | Okay |
| Vital | 58.33% | 8.33% |
| Okay | 8.33% | 25.00% |
| Irrelevant | 0% | 0% |

| Reference Importance | | |
|---|---|---|
| **Human / LLM** | Not Imp. | Imp. |
| Not Imp. | 40.2% | 9.8% |
| Imp. | 24.2% | 25.7% |

agreement score for labeling reference importance to compute Reference Coverage. Notably, each of these tasks require reasoning about complex academic literature and lengthy candidate related works sections. The observed agreement rates from our study provide promising results, validating the use of automated LLM-judges and metrics to assess complex generative research synthesis tasks.

For Organization, we see from the confusion matrix that the main point of human- and LLM-judges most often agree on pairwise comparisons, with strong disagreements (i.e., humans preferring the reference report and the LLM preferring the generated report or vice versa) are rare. Moreover, of the disagreements that occur between human- and LLM-judges, the LLM mis-judgments are relatively equally balanced between picking the Reference report and the Generated Report.

For Nugget Importance, we observe that in addition to the strong agreement rate observed from the confusion matrix, we also see that the human majority vote find all LLM-generated nuggets to be relevant, indicating hallucinations are rare. Moreover we see that the false postive and false negative rate of the LLM-judge are similar, and both rather small, less than $10\%$, once again indicating that severe LLM mis-labeling is rather rare.

Finally, for Reference Importance we observe an overall agreement score of $65.9\%$, and importantly false negative rate of $9.8\%$, i.e., when the LLM incorrectly labels a reference as important. The low false negative rate indicates a low likelihood of our Reference Coverage metric falsely penalizing systems. The larger off-diagonal mass ($24.2\%$) reflects under-labeling of essential references by the LLM, suggesting that our Reference Coverage scores provide a rather conservative metric, measuring "recall" of only a subset of all truly important references for each query.

## 7 CONCLUSION

In this work, we introduced DeepScholar-bench, a live dataset and holistic, automated evaluation framework designed to rigorously benchmark an emerging class of systems designed for generative research synthesis. By automatically sourcing queries from high-quality, recent ArXiv papers, our benchmark mitigates the risks of data staleness and training contamination, while offering a real research synthesis task. Moreover, DeepScholar-bench provides an automated evaluation to holistically measure three critical dimensions: retrieval quality, knowledge synthesis and verifiability. We further release DeepScholar-ref, a reference pipeline, which we find provides a strong baseline for generative research synthesis. Overall our systematic evaluation of prior open-source systems, search agents, OpenAI's DeepResearch and DeepScholar-ref demonstrates significant opportunities for future work, with no system surpassing a geometric mean of $31\%$ across all metrics. These results demonstrate both the difficulty of DeepScholar-bench and the exciting opportunity for further advancement in this space. We hope that DeepScholar-bench and DeepScholar-ref will support the development of more capable AI systems for generative research synthesis.

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

# 8 APPENDIX

| FIELD NAME | DESCRIPTION | EXAMPLE |
|---|---|---|
| arxiv_id | ArXiv paper ID | 2506.02838v1 |
| title | Paper title | TaxAgent: How Large Language Model Designs Fis... |
| authors | Comma-separated list of authors | Jizhou Wang, Xiaodan Fang, Lei Huang, Yongfeng... |
| abstract | Paper abstract from ArXiv | Economic inequality is a global challenge, int... |
| categories | ArXiv categories | cs.AI, econ.GN, q-fin.EC, I.2.11, I.6.5 |
| published_date | First Publication date | 2025-06-03T13:06:19+00:00 |
| clean_latex_related_works | Related Works section derived from LATEX of paper | \subsection{ Traditional Tax Systems}\nProgress ... |

| FIELD NAME | DESCRIPTION | EXAMPLE |
|---|---|---|
| **Parent Paper** | *Info about the paper in which the citation appears.* | |
| parent_paper_title | Title of the parent paper. | TaxAgent: How Large Language Model Designs Fis... |
| parent_paper_arxiv_id | ArXiv ID of the parent paper. | 2506.02838v1 |
| **Cited Paper** | *Details from the reference entry.* | |
| citation_shorthand | Citation key in the bibliography. | NBERw21340 |
| cited_paper_title | Title as listed in the reference list. | Effective Policy for Reducing Inequality? The ... |
| cited_paper_arxiv_link | ArXiv link if provided. | NaN |
| **Bibliographic Data** | *Metadata from the attached bibliography.* | |
| bib_paper_authors | Authors from external metadata. | Hoynes, Hilary W and Patel, Ankur J |
| bib_paper_year | Publication year | 2015 |
| bib_paper_month | Publication month | July |
| bib_paper_url | URL from bibliographic records. | http://www.nber.org/papers/w21340 |
| bib_paper_doi | DOI from external metadata. | 10.3386/w21340 |
| bib_paper_journal | Journal or series name. | NaN |
| original_title | Official title from bibliographic databases. | Effective Policy for Reducing Inequality? The ... |
| **Search Results (Verified)** | *Info obtained by searching online.* | |
| search_res_title | Title from the search result. | Effective Policy for Reducing Inequality? The ... |
| search_res_url | URL from the search result. | https://www.nber.org/papers/w21340 |
| search_res_content | Abstract snippet from the search result page. | We use a quasi-experiment approach, using vari... |

Figure 3: DeepScholar-bench dataset schema.

## 8.1 DEEPSCHOLAR-BENCH DATASET

We provide a detailed overview and schema of the DeepScholar-bench dataset in Figure 3. In addition, Table 5 includes the papers that are included in DeepScholar-June-2025.

## 8.2 OVERVIEW OF BASELINES AND EXPERIMENTAL SETUP

We briefly overview all of the baseline systems we evaluate, which includes recent state-of-the-art generative research systems as well as DeepScholar-ref on DeepScholar-bench. We benchmark open-source research systems, including DeepResearcher Zheng et al. (2025), STORM Shao et al. (2024) and OpenScholar Asai et al. (2024), search agents, with `Llama-4-Scout-17B-16E-Instruct` Meta (2025), `GPT-4.1-2025-04-14` OpenAI (2025b), `o3-2025-04-16` OpenAI (2025b), `Claude-opus-4-20250514` Anthropic (2025b), and `Gemini-2.5-pro` Gemini (2025b) models, `OpenAI's o3-deep-research` OpenAI (2025b), and DeepScholar-ref.

We report results using GPT-4.1-2025-04-14 (OpenAI, 2025b) as the judge for Nugget Coverage, and a GPT-4o-2024-08-06 (OpenAI, 2025b) judge for Organization, Relevance Rate, Reference Coverage, Citation Precision and Claim Coverage. We report the Organization score as a win rate including ties, we report the strict all score for Nugget Coverage, and we report Claim Coverage with a window size of $w = 1$. For all Retrieval Quality metrics, we consider the retrieved set of each given report as the set of any valid ArXiv links found within the report. To measure Document Importance, we use the OpenAlex (OpenAlex, 2025) API to recover citation information. For each metric, we report an average over all reports.

### 8.2.1 OPEN-SOURCE RESEARCH SYSTEMS

We evaluate three state-of-the-art open-source systems, DeepResearcher Zheng et al. (2025), STORM Shao et al. (2024) and OpenScholar Asai et al. (2024). For each, we run these systems using the `Llama-4-Scout-17B-16E-Instruct` model Meta (2025), which we serve with 4 A100 GPUs using vLLM Kwon et al. (2023).

DeepResearcher Zheng et al. (2025) leverages trained agents to navigate, browse and synthesize information from the web. To train an agent, this work uses end-to-end reinforcement learning and trains `Qwen2.5-7B-Instruct` Qwen et al. (2025). In our benchmarks,

Table 5: Title and ArXiv IDs of papers included in DeepScholar-June-2025.

| # | ArXiv ID | Title |
|---|----------|-------|
| 0 | 2506.02838v1 | TaxAgent: How Large Language Model Designs Fiscal Policy |
| 1 | 2506.02634v1 | KVCache Cache in the Wild: Characterizing and Optimizing KVCache Cache at a Large Cloud Provider |
| 2 | 2506.00958v1 | Speaking Beyond Language: A Large-Scale Multimodal Dataset for Learning Nonverbal Cues from Video-Grounded Dialogues |
| 3 | 2506.00832v1 | Counterfactual Activation Editing for Post-hoc Prosody and Mispronunciation Correction in TTS Models |
| 4 | 2506.00418v1 | Dual Debiasing for Noisy In-Context Learning for Text Generation |
| 5 | 2505.24754v1 | Don't Reinvent the Wheel: Efficient Instruction-Following Text Embedding based on Guided Space Transformation |
| 6 | 2505.24575v1 | NexusSum: Hierarchical LLM Agents for Long-Form Narrative Summarization |
| 7 | 2506.00085v1 | COSMIC: Generalized Refusal Direction Identification in LLM Activations |
| 8 | 2505.23996v1 | Is Your Model Fairly Certain? Uncertainty-Aware Fairness Evaluation for LLMs |
| 9 | 2505.23353v1 | Synthetic Generation and Latent Projection Denoising of Rim Lesions in Multiple Sclerosis |
| 10 | 2505.22757v1 | Pre-Training Curriculum for Multi-Token Prediction in Language Models |
| 11 | 2506.02853v1 | Learning Pyramid-structured Long-range Dependencies for 3D Human Pose Estimation |
| 12 | 2506.02547v1 | Probabilistic Online Event Downsampling |
| 13 | 2506.01071v1 | Aligned Contrastive Loss for Long-Tailed Recognition |
| 14 | 2506.01037v1 | Self-supervised ControlNet with Spatio-Temporal Mamba for Real-world Video Super-resolution |
| 15 | 2506.00434v1 | Efficient 3D Brain Tumor Segmentation with Axial-Coronal-Sagittal Embedding |
| 16 | 2506.00333v1 | Test-time Vocabulary Adaptation for Language-driven Object Detection |
| 17 | 2505.24443v1 | Diversify and Conquer: Open-set Disagreement for Robust Semi-supervised Learning with Outliers |
| 18 | 2505.24334v1 | KairosAD: A SAM-Based Model for Industrial Anomaly Detection on Embedded Devices |
| 19 | 2505.23290v1 | Wav2Sem: Plug-and-Play Audio Semantic Decoupling for 3D Speech-Driven Facial Animation |
| 20 | 2505.23180v1 | Proximal Algorithm Unrolling: Flexible and Efficient Reconstruction Networks for Single-Pixel Imaging |
| 21 | 2505.22616v1 | PS4PRO: Pixel-to-pixel Supervision for Photorealistic Rendering and Optimization |
| 22 | 2505.22458v1 | Universal Domain Adaptation for Semantic Segmentation |
| 23 | 2505.22427v1 | RC-AutoCalib: An End-to-End Radar-Camera Automatic Calibration Network |
| 24 | 2505.22167v1 | Q-VDiT: Towards Accurate Quantization and Distillation of Video-Generation Diffusion Transformers |
| 25 | 2505.22552v1 | ClaimPKG: Enhancing Claim Verification via Pseudo-Subgraph Generation with Lightweight Specialized LLM |
| 26 | 2504.21752v1 | VDDP: Verifiable Distributed Differential Privacy under the Client-Server-Verifier Setup |
| 27 | 2504.21282v1 | Birdie: Natural Language-Driven Table Discovery Using Differentiable Search Index |
| 28 | 2504.17448v1 | CHASe: Client Heterogeneity-Aware Data Selection for Effective Federated Active Learning |
| 29 | 2504.14861v1 | Stitching Inner Product and Euclidean Metrics for Topology-aware Maximum Inner Product Search |
| 30 | 2504.06975v1 | AWDIT: An Optimal Weak Database Isolation Tester |
| 31 | 2506.01833v1 | SPACE: Your Genomic Profile Predictor is a Powerful DNA Foundation Model |
| 32 | 2506.00382v1 | Spectral Insights into Data-Oblivious Critical Layers in Large Language Models |
| 33 | 2506.00205v1 | Unlocking the Power of Rehearsal in Continual Learning: A Theoretical Perspective |
| 34 | 2505.24835v1 | Timing is important: Risk-aware Fund Allocation based on Time-Series Forecasting |
| 35 | 2505.24203v1 | Aligning Protein Conformation Ensemble Generation with Physical Feedback |
| 36 | 2506.02847v1 | CLONE: Customizing LLMs for Efficient Latency-Aware Inference at the Edge |
| 37 | 2505.22194v1 | Refining Datapath for Microscaling ViTs |
| 38 | 2505.11554v1 | Multi-Objective Memory Bandwidth Regulation and Cache Partitioning for Multicore Real-Time Systems |
| 39 | 2505.08071v1 | NMP-PaK: Near-Memory Processing Acceleration of Scalable De Novo Genome Assembly |
| 40 | 2504.06211v1 | Need for zkSpeed: Accelerating HyperPlonk for Zero-Knowledge Proofs |
| 41 | 2504.19283v1 | Efficient Serverless Cold Start: Reducing Library Loading Overhead by Profile-guided Optimization |
| 42 | 2504.11007v1 | Kubernetes in the Cloud vs. Bare Metal: A Comparative Study of Network Costs |
| 43 | 2504.09307v1 | Lumos: Efficient Performance Modeling and Estimation for Large-scale LLM Training |
| 44 | 2506.02750v1 | Learning Binarized Representations with Pseudo-positive Sample Enhancement for Efficient Graph Collaborative Filtering |
| 45 | 2505.23452v1 | What About Emotions? Guiding Fine-Grained Emotion Extraction from Mobile App Reviews |
| 46 | 2505.21811v1 | Revisiting Self-attention for Cross-domain Sequential Recommendation |
| 47 | 2505.20227v1 | Measure Domain's Gap: A Similar Domain Selection Principle for Multi-Domain Recommendation |
| 48 | 2505.19356v1 | Optimized Text Embedding Models and Benchmarks for Amharic Passage Retrieval |
| 49 | 2505.19307v1 | Aligning Web Query Generation with Ranking Objectives via Direct Preference Optimization |
| 50 | 2505.17507v1 | Benchmarking Recommendation, Classification, and Tracing Based on Hugging Face Knowledge Graph |
| 51 | 2505.12791v1 | Unlearning for Federated Online Learning to Rank: A Reproducibility Study |
| 52 | 2505.07166v1 | Pre-training vs. Fine-tuning: A Reproducibility Study on Dense Retrieval Knowledge Acquisition |
| 53 | 2505.03484v1 | STAR-Rec: Making Peace with Length Variance and Pattern Diversity in Sequential Recommendation |
| 54 | 2505.00552v1 | Graph Spectral Filtering with Chebyshev Interpolation for Recommendation |
| 55 | 2504.20458v1 | Search-Based Interaction For Conversation Recommendation via Generative Reward Model Based Simulated User |
| 56 | 2504.18383v1 | Bridge the Domains: Large Language Models Enhanced Cross-domain Sequential Recommendation |
| 57 | 2504.17519v1 | Replication and Exploration of Generative Retrieval over Dynamic Corpora |
| 58 | 2504.15849v1 | NLCTables: A Dataset for Marrying Natural Language Conditions with Table Discovery |
| 59 | 2504.14991v1 | Understanding Accuracy-Fairness Trade-offs in Re-ranking through Elasticity in Economics |
| 60 | 2504.14243v1 | Unconstrained Monotonic Calibration of Predictions in Deep Ranking Systems |
| 61 | 2504.12900v1 | FashionDPO:Fine-tune Fashion Outfit Generation Model using Direct Preference Optimization |
| 62 | 2504.09935v1 | Constrained Auto-Regressive Decoding Constrains Generative Retrieval |

we evaluate DeepResearcher using both the released, trained model from the authors, and using `Llama-4-Scout-17B-16E-Instruct` model Meta (2025) as the core LLM. We report the better performing baseline of these two, which we find in our experiments to be the `Llama-4-Scout-17B-16E-Instruct` backbone.

STORM Shao et al. (2024) studies the problem of how to apply LLMs to write grounded, organized long-form articles (e.g., Wikipedia articles) from scratch. The system involves a pre-writing stage that discovers diverse research perspectives on a topic by stimulating conversations between multiple agents and leveraging web documents.

OpenScholar Asai et al. (2024) builds a specialized retrieval-augmented LLM system for literature synthesis and scientific queries. This method includes a trained retriever from the pre-indexed peS2o Soldaini et al. (2024) corpus, consisting of 45 million open-access academic papers up until October 2024, as an initial retrieval source before using web search. In our experiments, we benchmark the system using this pre-indexed corpus and limit web search to the ArXiv API.

### 8.2.2 SEARCH AGENTS

We evaluate the following models: `Llama-4-Scout-17B-16E-Instruct` Meta (2025), `GPT-4.1-2025-04-14` OpenAI (2025b), `o3-2025-04-16` OpenAI (2025b), `Claude-opus-4-20250514` Anthropic (2025b), and `Gemini-2.5-pro` Gemini (2025b). We augment each with search capabilities to ArXiv arxiv (2025), and use the popular ODS framework Alzubi et al. (2025) to allow the LLM to make tool calls to the search API.

### 8.2.3 COMMERCIAL SYSTEMS.

We focus our evaluation of commercial generative research synthesis systems on OpenAI's o3-deepresearch OpenAI (2025b), which provides a public API allowing for our evaluation.

### 8.2.4 DEEPSCHOLAR-REF

Similar to our evaluation of search agent we evaluate DeepScholar-ref with the following models: `Llama-4-Scout-17B-16E-Instruct` Meta (2025), `GPT-4.1-2025-04-14` OpenAI (2025b), `o3-2025-04-16` OpenAI (2025b), `Claude-opus-4-20250514` Anthropic (2025b), and `Gemini-2.5-pro` Gemini (2025b). For each of these baselines, we also use the same or a weaker model, either Llama-4 or GPT-4.1, to perform semantic filtering and top-k operators. We limit the method to two rounds of search, each with at most 2 queries.

## 8.3 ADDITIONAL EXPERIMENTAL RESULTS

### 8.3.1 METADATA STATISTICS

We provide metadata statistics characterizing the generated reports of each benchmarked method in Table 6, as well as statistics related to our evaluation metrics in Table 7.

### 8.3.2 RESULTS ON DEEPSCHOLAR-NOV-2025

To study how our benchmark behaves under both domain and temporal shifts in the live arXiv API, we instantiate a second benchmark slice, in addition to DeepScholar-June-2025 (Section 3 which is built from 63 Computer Science papers on arXiv to study domain coverage and robustness of DeepScholar-bench.

DEEPSCHOLAR-NOV-2025 contains 200 queries sampled from more than 75 distinct arXiv subject areas, spanning Computer Science, Physics, Quantitative Biology, Economics, and Quantitative Finance. We evaluate a subset of high-performing systems to check whether our conclusions remain stable: DeepScholar-ref instantiated with Llama-4-Scout-17B-16E and with GPT-4.1+o3, and the Search Agent baseline instantiated with the same model configurations.

Table 8 reports the resulting scores across all seven metrics. Overall, we observe patterns that are consistent with our main results on DeepScholar-June-2025 (Table 2). For both DeepScholar-ref and the Search Agent baseline, the o3-based variants substantially outperform their Llama-4-Scout

Table 6: Report Statistics.

| | Report Length | | | Citations | | Cost | |
|---|---|---|---|---|---|---|---|
| | Chars | Words | Sentences | # Unique Refs | # Inline Citations | Latency (s) | Dollar Cost (USD) |
| *Human-written Exemplars* | | | | | | | |
| Human-written Exemplars | 4381 | 497 | 28 | 23 | 27 | N/A | N/A |
| *Open Source Research Systems* | | | | | | | |
| DeepResearcher (Llama-4) | 2573 | 319 | 35 | 8 | 7 | 31 | 0.00 |
| STORM (Llama-4) | 2766 | 381 | 31 | 18 | 21 | 162 | 0.00 |
| OpenScholar (Llama-4) | 3513 | 483 | 26 | 9 | 19 | 87 | 0.00 |
| *Search Agents* | | | | | | | |
| Search Agent (Llama-4) | 1968 | 258 | 16 | 9 | 5 | 20 | 0.00 |
| Search Agent (GPT-4.1) | 3168 | 404 | 16 | 10 | 61 | 39 | 0.07 |
| Search Agent (o3) | 3844 | 501 | 24 | 11 | 16 | 263 | 0.15 |
| Search Agent (Claude) | 3977 | 499 | 27 | 13 | 8 | 147 | 1.36 |
| Search Agent (Gemini) | 2810 | 395 | 19 | 6 | 8 | 442 | 0.11 |
| *Commercial Systems* | | | | | | | |
| OpenAI DeepResearch | 6577 | 864 | 74 | 17 | 6 | 630 | 5.02 |
| *DeepScholar Reference Pipeline* | | | | | | | |
| DeepScholar-ref (Llama-4) | 3499 | 360 | 53 | 19 | 35 | 313 | 0.00 |
| DeepScholar-ref (GPT-4.1) | 7863 | 735 | 115 | 20 | 89 | 234 | 1.66 |
| DeepScholar-ref (GPT-4.1, o3) | 5726 | 617 | 70 | 17 | 42 | 276 | 1.15 |
| DeepScholar-ref (GPT-4.1, Claude) | 5855 | 618 | 72 | 17 | 40 | 334 | 1.23 |
| DeepScholar-ref (GPT-4.1, Gemini) | 5623 | 570 | 86 | 24 | 63 | 349 | 1.29 |

Table 7: Statistics Related to Evaluation Metrics.

| | Avg. value over human-written exemplars | Relevant Metric |
|---|---|---|
| # Important References from ArXiv.org | 11.47 | Ref. Cov. |
| Median number of citations per reference from ArXiv.org | 647.5 | Doc. Imp. |

counterparts on Organization, Nugget Coverage, Relevance Rate, Citation Precision, and Claim Coverage, while Reference Coverage and Document Importance remain broadly low across baselines. These trends mirror the relative ranking and qualitative gaps seen in our main benchmark slice, suggesting that our conclusions from DeepScholar-June-2025 generalize robust beyond queries related to Computer Science.

We emphasize that DeepScholar-Bench is defined by an automated data-curation and evaluation pipeline rather than a single fixed dataset. Instantiating new slices such as DEEPSCHOLAR-NOV-2025 only requires specifying a set of query papers and a date range; the pipeline then automatically constructs the corresponding benchmark and produces scores under the same evaluation protocol. This design allows practitioners to easily create additional domain- or time-specific evaluations while remaining comparable to our core results.

### 8.3.3 ABLATION STUDY: UNDERSTANDING THE PERFORMANCE IMPACT OF THE CHOSEN QUERY

Our main experiments use the paper abstract as the query description $d$ (Section 3.2). A natural question is whether our conclusions depend on this particular choice of query formulation. To assess this, we run an ablation in which we replace the abstract with two alternative, realistic user queries: (i) a two-sentence summary of the paper's key idea (KEY IDEA) and (ii) a single research question describing the paper's main goal (RQ). For each paper, we prompt an LLM to convert the abstract into these two alternative query formulations. We then re-run the full benchmark for each query version and compute system-level scores for all metrics across our main baselines.

Table 9 reports Pearson correlations between system-level scores obtained under different query formulations (rows) for each metric (columns) with statistical significance testing based on a permutation-based paired test ($p<0.05$). Overall, we observe very strong agreement across query types: correlations are typically above 0.95 for Organization, Nugget Coverage, Reference Coverage, Document Importance, and Claim Coverage, and above 0.77 for Coverage Relevance Rate in

Table 8: Performance of selected systems on DEEPSCHOLAR-NOV-2025 (200 queries across >75 arXiv subject areas). We report the same metrics as in Table 2.

| Model | Org. | Nug. Cov. | Rel. Rate | Ref Cov. | Doc Imp. | Cite-P | Claim Cov. |
|---|---|---|---|---|---|---|---|
| DeepScholar-ref (Llama-4-Scout) | 0.120 | 0.358 | 0.395 | 0.072 | 0.082 | 0.178 | 0.581 |
| DeepScholar-ref (GPT-4.1 + o3) | 0.578 | 0.479 | 0.568 | 0.087 | 0.056 | 0.563 | 0.578 |
| Search Agent (Llama-4-Scout) | 0.108 | 0.252 | 0.179 | 0.034 | 0.079 | 0.189 | 0.314 |
| Search Agent (o3) | 0.608 | 0.480 | 0.623 | 0.078 | 0.034 | 0.475 | 0.452 |

Table 9: Pearson correlations between system-level scores under different query formulations (abstract, KEY IDEA, and RQ). Each entry is computed over the scores of our main baselines. Correlations marked with * are statistically significant under a permutation-based paired test with threshold $p<0.05$.

| Query pair | Org. | Nug. | Cov. Rel. Rate | Ref Cov | Doc Imp | Cite-P | Claim Cov ($w$=1) |
|---|---|---|---|---|---|---|---|
| Abstract vs. Key Idea | 0.997* | 0.980* | 0.979* | 0.992* | 0.992 | 0.589 | 0.841* |
| Abstract vs. RQ | 0.988* | 0.979* | 0.772* | 0.951* | 0.707 | 0.766* | 0.877* |
| Key Idea vs. RQ | 0.991* | 0.983* | 0.836* | 0.958* | 0.766* | 0.942* | 0.981* |

all cases. All correlations between KEY IDEA and RQ are statistically significant, indicating that more natural, user-facing query formulations yield highly consistent system rankings.

The main sensitivity to query formulation arises for Document Importance and, to a lesser extent, Citation Precision. The abstract vs. Key Idea and abstract vs. RQ correlations on Document Importance, as well as the abstract vs. Key Idea correlation on Citation Precision, are not statistically significant, despite having large magnitudes (e.g., $r=0.992$ for Document Importance). This suggests that these two metrics are most sensitive to how the query is phrased, likely because small changes in the query can shift which highly-cited papers are retrieved and cited. In contrast, the remaining metrics exhibit high and statistically significant correlations across all query pairs. Taken together, these results indicate that our benchmark conclusions are broadly robust to reasonable variations in query formulation, with only modest sensitivity in how document-level importance and citation precision are expressed across different query types.

### 8.3.4 HUMAN EVALUATION DETAILS

To assess whether LLM-based metrics reflect human expert judgments, we conduct a human evaluation with 11 annotators, all Computer Science PhD students from four research universities across North America. In total, we collect over 300 human annotations aimed to assess the agreement between humans and LLMs in order to validate the LLM-based judges introduced by our automated evaluation for assessing knowledge synthesis and retrieval quality. We describe our setup in detail below.

**Knowledge Synthesis.** For Organization & Coherency, we sample queries and show annotators the human-written related work section alongside a system-generated report. For each pair, annotators indicate whether they prefer the system report, prefer the human-written exemplar, or consider them similarly organized. These labels are used to evaluate the pairwise comparison outcomes underlying our Organization metric. For Nugget Coverage, we first generate information nuggets from the human-written related work section. Annotators are then shown individual nuggets and asked to judge whether each nugget is *vital* for understanding the paper, *okay*, or *irrelevant*. These nugget-importance labels determine which nuggets are treated as essential when computing nugget coverage scores.

**Reference coverage and essential citations.** To ground our Reference Coverage metric in human judgments, we ask each annotator to select a high quality paper whose related work section they are comfortable with. For that paper, the annotator identifies at least six references they consider *important* (i.e., references that should appear in a good related work section) and at least six references they consider *not important* (i.e., references that could be omitted or substituted without

Table 10: Manual Validation of LLM-based Evaluation

| Evaluation Metric | LLM-Classified Labels | Human-Agreement Score with LLM |
|---|---|---|
| Organization | Pairwise Comparison (Lose / Tie / Win) | 78% |
| Nugget Coverage | Nugget Importance (Vital / Non-vital) | 72% |
| Nugget Coverage | Nugget Coverage (Supported / Partially Supp. / Not Supp.) | 70% |
| Retrieval Relevance Rate | Graded Relevance (0/1/2) | 70% |
| Reference Coverage | Reference Importance (Not Imp./ Imp.) | 82% |
| Document Importance | N/A | N/A |
| Citation Precision | Entailment (Entailed / Not Entailed) | 80% |
| Claim Coverage | Entailment (Entailed / Not Entailed) | 80% |

Table 11: Pearson correlations between system-level scores produced by different LLM judges across five baselines for each metric. Statistical significance is assessed using a permutation-based paired t-test with threshold $p<0.05$; correlations with $p<0.05$ are marked with $^*$.

| Judge pair | Org. | Nug. | Cov. Rel. Rate | Ref. Cov. | Cite-P | Claim Cov ($w=1$) |
|---|---|---|---|---|---|---|
| GPT-4o vs. Llama-4 | 0.985$^*$ | 0.413 | 0.941 | 0.975$^*$ | 0.817$^*$ | 0.958$^*$ |
| GPT-4o vs. DeepSeek | 0.966$^*$ | 0.920$^*$ | 0.980$^*$ | 0.990$^*$ | 0.843$^*$ | 0.996$^*$ |
| DeepSeek vs. Llama-4 | 0.991$^*$ | 0.668 | 0.942$^*$ | 0.994$^*$ | 0.986 | 0.963$^*$ |

harming the quality of the section). These labels form gold sets of important versus non-important references, which we use both to evaluate Reference Coverage and to validate our LLM-based importance labels. We then compare human majority labels to predictions from our LLM-judge over more than 130 blind reference-level annotations. The resulting confusion matrix (rows = human labels, columns = LLM predictions) is shown in Table 4.

### 8.3.5 MANUAL VALIDATION OF LLM-BASED EVALUATION

We study the alignment between our LLM-based evaluation and human judgments to assess the effectiveness of our automated metrics. Overall, we find that each of the metrics we introduce for the DeepScholar-bench task exhibit high agreement between LLM-based judgments and human annotations. We collect over 400 human annotations, and Table 10 shows the agreement score between human and LLMs for each LLM classification task associated with each automated metrics. The results demonstrate above 70% agreement scores across each. We additionally compute the nugget precision and grounded-ness scores, following prior work (Thakur et al., 2025), observing scores of .83 and 1.0 respectively, demonstrating that the generated nuggets are accurate and do not contain hallucinations.

### 8.3.6 AGREEMENT BETWEEN DIFFERENT LLM JUDGES

To test the robustness of our evaluation to the choice of LLM judge, we repeat all experiments with three different judges: GPT-4o, Llama-4-17b-16e-Instruct, and DeepSeek-R1-Distill-Qwen-32B. For each pair of judges, we compute the Pearson correlation between system-level scores across five baselines (DeepScholar-ref (GPT-4.1, Claude), DeepScholar-ref (Llama-4), Search Agent (Claude), Search Agent (Llama-4), and OpenAI DeepResearch for all metrics except document importance, which is not LLM-based. We then run perturbation-based permutation tests to assess whether these correlations are significantly greater than zero. The resulting correlations and significance markers are shown in Table 11.

Correlations are generally very high (often above 0.9). Notably GPT-4o vs. DeepSeek shows statistically significant correlations across all the metrics at the $p < 0.05$ level, indicating that these two judges high agreement. Most metrics also remain significantly correlated, but we observe the main sensitivity to the choice of judge on nugget coverage (non-significant for both GPT-4o vs. Llama-4-17b-16e-Instruct and DeepSeek-R1-Distill-Qwen-32B vs. Llama-4-17b-16e-Instruct). In addition, relevance rate shows non-statistically significant correlation for GPT-4o vs. Llama-4-17b-16e-Instruct and similarly Citation Precision for DeepSeek-R1-Distill-Qwen-32B vs. Llama-4-17b-16e-Instruct. It is worth mentioning that even in these cases, the correlations remain large (all above 0.41), suggesting that discrepancies are not severe.

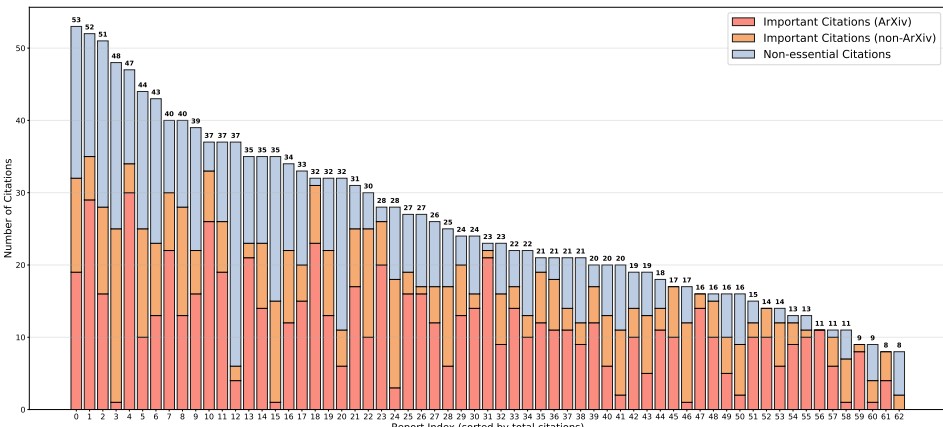

Figure 4: Citation importance breakdown in DeepScholar-Bench. Each bar corresponds to a single human exemplar related work section, sorted by the total number of citations. Bars are color-coded to indicate *important ArXiv citations* (red), *important non-ArXiv citations* (orange), and *non-essential citations* (blue).

## 8.4 EVALUATION DETAILS

Here we provide the detailed implementation and prompts for the evaluation metrics used in DeepScholar-Bench. In the following, we also run some ablation studies on different metrics.

- **Knowledge Synthesis – Organization:** Box 1 shows the prompt used to compare system-generated reports with the human-written exemplar, yielding the win-rate of each system based on organization and coherence.

- **Knowledge Synthesis – Nugget Coverage:** For this metric we follow the nugget-based evaluation prompts from prior work Pradeep et al. (2025), which extract essential facts from the exemplar and check their presence in the generated report.

- **Retrieval Quality – Document Importance:** We select important references using the LOTUS program shown in Figure 5.

- **Retrieval Quality – Relevance Rate:** Following the Cranfield model of relevance, in Box 2 we prompt the model for assigning graded relevance scores (0–2) to each retrieved source.

- **Verifiability – Citation Precision and Claim Coverage:** Following prior work Gao et al. (2023); Liu et al. (2023), we check entailment of claims with respect to cited sources. using the prompt shown in Box 3 for both Citation Precision and Claim Coverage as explained in Section 4.3.

We note that Reference Coverage and Document Importance are computed deterministically as explained in Section 4.2: the former by comparing the retrieved set against the important references in the human-written exemplar, and the latter by normalizing the number of citations of the reference papers.

### 8.4.1 REFERENCE COVERAGE

Figure 4 illustrates the distribution of important citations across the human exemplar reports in DeepScholar-Bench. For each exemplar, we used the LOTUS program shown in Figure 5 to identify which citations are *important* and therefore essential to include in a high-quality related work section. We then separate these important citations into two groups: those that appear on ArXiv (shown in red) and those that do not (shown in orange). The blue portion of each bar corresponds to *non-essential* citations, as determined by the same Lotus-based procedure.

The plot highlights two consistent trends. First, many exemplar related work sections contain a large number of non-essential citations. While such references may be useful for narrative flow or broader

context, they are not indispensable. Non-essential citations can be somewhat subjective, depending on how authors choose to frame the story of their paper. In contrast, the important citations represent the "must-have" references i.e., the foundational works in the field that are necessary for situating the contribution. Second, we observe that the red segments (important ArXiv citations) are well distributed across exemplars, indicating that ArXiv is a reliable and sufficiently broad source for recovering many of the essential references.

```
query_in = "Carefully read the {title}, {abstract} and {
    related_work_section} of an academic paper. \
    Then consider the cited paper in question, given the title {
    cited_paper_title}, the {cited_paper_authors} and a snippet of its
    content, {cited_paper_content}.\
    Is the cited paper in question an essential reference?\
    An essential reference reflects a key, notable prior work that
    provides key information, which a good related works section for
    this paper must include.\
    A non-essential reference is one that is not essential to the
    related work section of this paper and could be omitted or
    substituted with a different reference.\
    a non-essential reference may be a relevant reference that reflects
    an important topic area, but the particular reference could be
    omitted or substituted with a different related work.\
    Alternatively, a non-essential reference may be a tangential
    reference, an unimportant reference.\a non-essential reference may
    be a relevant reference that reflects an important topic area, but
    the particular reference could be omitted or substituted with a
    different related work."

res = citations_df.sem_filter(query_in, return_all=True, strategy=lotus.
    types.ReasoningStrategy.ZS_COT)
```

Figure 5: LOTUS program for Finding Important References

### 8.4.2 Ablation Study on Verifiability

In the main paper (Section 4.3, we reported verifiability metrics results using a sliding window of size $w = 1$ when computing claim coverage. That is, for each claim sentence, we considered a citation to be valid if any of the references in the same sentence or within one sentence before or after sufficiently supported the claim. Here, we extend this analysis to study the effect of varying the window size. Specifically, we report the citation coverage achieved by different systems when the window size ranges from $w = 0$ (same-sentence only) up to $w = 5$ (five sentences before or after).

As shown in Figure 6, increasing the window size consistently improves citation coverage across all baselines. This is expected: the larger the window, the higher the probability that one of the cited references in the $[-w, +w]$ neighborhood of a claim provides sufficient support. However, we also note that very large window sizes are less desirable in practice, as they often correspond to references being far from the claims they are intended to support, reducing readability and making it harder for readers to verify the connection between claims and citations. Moreover, from Table 6, we see that real academic writing tends to be densely cited, with at least one citation on average per sentence in the human exemplars. Overall, the results of our ablation study highlight the trade-off between stricter precision ($w = 0$) and more lenient recall-oriented settings ($w \geq 1$).

### 8.4.3 Document Importance Across Human Exemplars

In this section, we illustrate the distribution of document importance, measured by the number of citations of references in the human-written exemplars in DeepScholar-Bench. Figure 7 reports

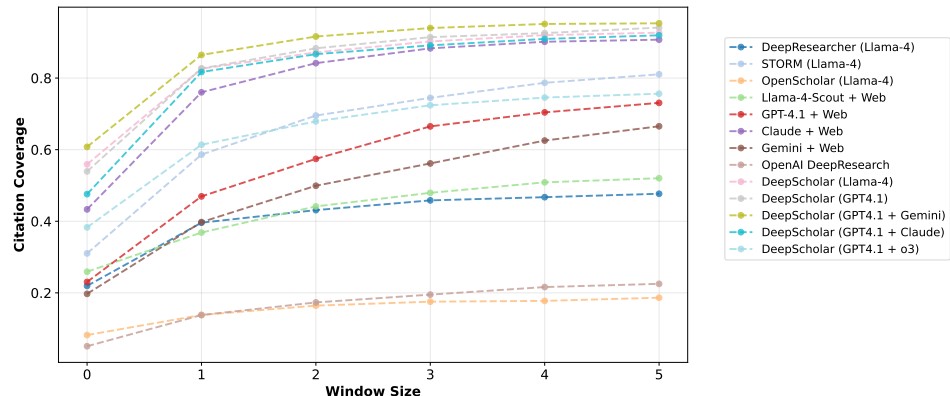

Figure 6: Ablation study on citation coverage with different window sizes. For each claim, we measure whether any citation within a sliding window of $[-w, +w]$ sentences supports it.

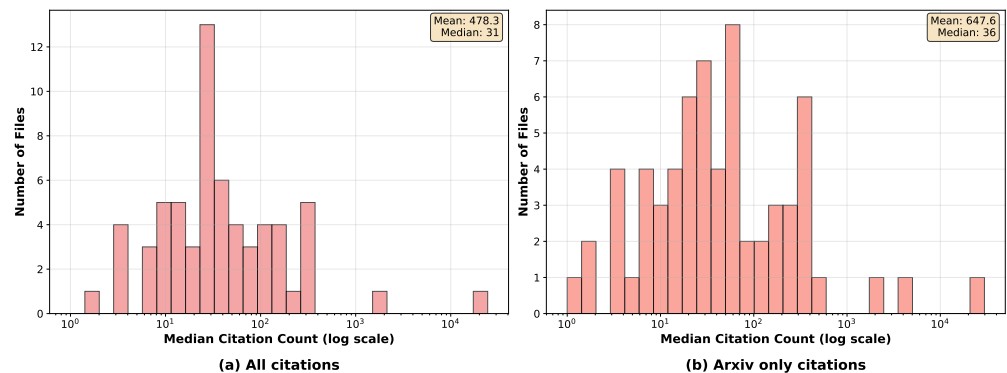

Figure 7: Distribution of citation counts (Document Importance) for references in human-written exemplars. Figure (a) shows all references, while Panel (b) restricts to ArXiv references only. Citation counts are plotted on a logarithmic scale.

two histograms: (a) the distribution of citation counts across all references, and (b) the distribution restricted to references that appear on ArXiv. We plot the logarithm of citation counts, with values obtained from the OpenAlex API OpenAlex (2025), an open and widely used scholarly database that provides citation-level metadata. While citation counts in OpenAlex may not exactly match those from other sources such as Google Scholar, the relative counts are consistent, making it a reliable open-source alternative.

As shown in Figure 7, the distribution is highly skewed due to a small number of papers with exceptionally large citation counts (e.g., over 10k citations). These outliers inflate the mean citation values, resulting in relatively high averages compared to typical references (478.3 citations across all references and 647.6 for ArXiv-only references). In contrast, the median values are lower (31 for all references and 36 for ArXiv-only). This skew highlights the challenge of using citation counts as a proxy for importance, as the median citation count of references, among different human-written exemplars exhibits high variance.

## 8.5 EXTENDED DESCRIPTION OF BASELINES

We provide an extended description of each benchmarked method, including relevant implementation details, and parameters used in our evaluation.

1350
1351
1352
1353
1354
1355
1356
1357
1358
1359
1360
1361
1362
1363
1364
1365
1366
1367
1368
1369
1370
1371
1372
1373
1374
1375

---

**Box 1: Prompt for Knowledge Synthesis- Organization**

You are an intelligent, rigorous, and fair evaluator of scholarly writing quality and relevance. You will receive the title and abstract of a research paper, together with two candidate related-work sections (A and B) written for that paper. Do not consider the formatting of the text (e.g., LaTeX, markdown, etc.). Only consider the content.

Task: Decide which section—A or B—exhibits better organization and coherence.
How to judge (organization only) Ignore breadth of coverage, citation accuracy, and analytic depth. Assess:
Logical structure – Clear introduction, grouping of related themes, and smooth progression of ideas.
Paragraph cohesion – Each paragraph develops a single topic and flows naturally to the next.
Clarity & readability – Minimal redundancy or contradictions; transitions guide the reader.
Signposting – Helpful headings, topic sentences, or discourse markers (if provided).

Pick the section that is easier to follow and better structured—no ties.

### Paper under assessment: `[TITLE + ABSTRACT GO HERE]`
### Candidate related-work section A `[RELATED WORK A TEXT GOES HERE]`
### Candidate related-work section B `[RELATED WORK B TEXT GOES HERE]`

Output your answer as a JSON dictionary in the following format:
`{"decision": "A" or "B", "explanation": "One sentence clearly explaining the key differences between the two options and why the selected one is preferred."}`
Only output the dictionary, do not output any other text.

---

1376
1377
1378
1379
1380
1381
1382
1383
1384
1385
1386
1387
1388
1389
1390
1391
1392
1393
1394
1395
1396
1397
1398
1399
1400
1401
1402
1403

---

**Box 2: Prompt for Reference Relevance Judgment**

You are an intelligent, rigorous, and fair evaluator of scholarly writing quality and citation relevance. You will receive the title and abstract of a research paper under assessment, the ground-truth related-work section written by human experts, and the title and abstract of a candidate reference paper. Do not consider formatting (e.g., LaTeX, markdown, etc.). Only consider the content.

Task: Determine whether the candidate reference paper is relevant to the related-work section.
How to judge • Consider the main research topic and themes described in the related-work section.
• If the reference discusses similar ideas, prior work, or background, mark it as relevant (1).
• If the reference is off-topic or unrelated in scope, mark it as not relevant (0).
• Remember: You are only seeing the title and abstract of the reference, so the full content might be more relevant than it appears.

### Paper under assessment: `[PAPER TITLE GOES HERE]` `[PAPER ABSTRACT GOES HERE]`
### Ground-truth related-work section: `[RELATED WORK TEXT GOES HERE]`
### Candidate reference paper: `[REFERENCE TITLE GOES HERE]` `[REFERENCE ABSTRACT GOES HERE]`

Return only the score in this format:
`### final score: <0 or 1>`

---

---

**Box 3: Prompt for Attribution Validation**

You are an intelligent and fair evaluator. Your task is to verify whether a given reference can support the provided claim.

Task: Given a claim and its associated set of references, determine whether the references sufficiently support all aspects of the claim.
### CLAIM: [CLAIM TEXT GOES HERE]
### REFERENCES: [REFERENCE TEXT GOES HERE]

Judgment Criteria: • If the references support the claim, return `1`.
• If the references do not support the claim, return `0`.
• Do not explain your answer or include any additional commentary.

Output Format:
`Answer: 1` or `Answer: 0`

---

### 8.5.1 DEEPRESEARCHER

The DeepResearcher pipeline follows a structured tool-augmented reasoning framework designed for iterative web-based information retrieval. The system mandates explicit reasoning before any tool invocation, with reasoning encapsulated in `<think>` tags to ensure interpretability and control. After reasoning, the model generates a JSON-formatted request specifying the "web search" tool and its query. These queries are executed via the Lotus Search API, which we replaced with an ArXiv-specific search interface to provide a controlled retrieval API for our evaluation. Retrieved results are returned in a structured format containing the title, URL, and snippet, and are stored in memory for reference across subsequent reasoning steps. This iterative process continues until the model determines that sufficient evidence has been gathered, after which a synthesized final response is produced.

For our experiments, we used `Llama-4-Scout-17B-16E-Instruct` as the base model, replacing the originally proposed DeepResearcher-7b, since it demonstrated consistently better retrieval-augmented reasoning performance in our experiments. The prompt was slightly modified to align with LLama-4 prompt style as detailed in Box 4. The retrieval depth was set to 10 sources per query, which is the default in the system and provides a balanced trade-off between coverage and efficiency. We restricted each query to a single rollout with a maximum of 10 steps, following the DeepResearcher defaults; this limit is generous as most rollouts converge in fewer than three steps, but it ensures the system has headroom for more complex queries. The default web search API was replaced with ArXiv search to comply with our benchmark settings.

listings

### 8.5.2 OPENSCHOLAR

The OpenScholar pipeline follows a four-stage process: initial retrieval, response and feedback generation, iterative refinement, and citation verification. In the first stage, text segments are retrieved from a fixed index using a contriever model, which encodes texts and retrieves passages based on semantic similarity. These passages are reranked and used to generate an initial draft response, where citations are aligned with the supporting passages. The second stage introduces feedback generation, where the model produces up to three feedback statements highlighting potential improvements in the draft, such as missing content or organization issues; if additional evidence is required, retrieval queries are issued. The third stage iteratively refines the response by conditioning on the previous draft, retrieved passages, and newly added evidence, yielding improved responses at each step until feedback has been fully incorporated. Finally, citation verification ensures that all citation-worthy statements are adequately grounded in the retrieved sources, inserting additional citations where necessary without removing content.

For consistency with other baselines, we employ the `Llama-4-Scout-17B-16E-Instruct` model for generation. The retrieval pipeline initially collects 100 text segments from

---

**Box 4: Revised DeepResearcher System Prompt optimized to work with Llama-4-Scout-17B-16E-Instruct**

## Background information
* Today is `{strftime("%Y-%m-%d", gmtime())}`
* You are Deep AI Research Assistant The question I give you is a complex question that requires a *deep research* to answer. I will provide you with two tools to help you answer the question:
* A web search tool to help you perform google search. Tool call format:

```
{{"name": "web_search",
"arguments": {{"query": ["<query1>","<query2>","<query3>"]}}}}
```

* A webpage browsing tool to help you get new page content. Tool call format:

```
{{"name": "browse_webpage",
"arguments": {{"url_list": ["<url1>","<url2>","<url3>"]}}}}
```

You don't have to answer the question now, but you should first think about the research plan or what to search next.
Your output format should be one of the following two formats:

```
<think>
YOUR THINKING PROCESS
</think>
<answer>
YOUR ANSWER AFTER GETTING ENOUGH INFORMATION
</answer>
```

or

```
<think>
YOUR THINKING PROCESS
</think>
<tool_call>
YOUR TOOL CALL WITH CORRECT FORMAT
</tool_call>
```

You should always follow the above two formats strictly. You will be heavily penalized if you do not follow the format strictly. Only output the final answer (in words, numbers or phrase) inside the <answer></answer> tag, without any explanations or extra information. If this is a yes-or-no question, you should only answer yes or no.

---

`peS2o_v3` using the default `pes2o_contriever`[1] model. The reranker used is `OpenScholar_Reranker`[2], also kept at its default setting. To align parameterization across baselines, we increase the number of sources used in generation (`top_n`) from 10 to 30. Furthermore, the default search API is replaced with the arXiv API, to provide a controlled retrieval corpus and API in our experiments.

### 8.5.3 SEARCH AGENT

Search Agents are implemented using each LLM API with search API access. We implement a ReAct Agent, instantiated from the `smolagents.ToolCallingAgent`, with a system prompt based on the open-source OpenDeepSearch (ODS) framework designed for deep web search and retrieval, along with the search agent as an external tool. At each reasoning step, the ReAct agent can either invoke the search agent through a `web_search` action or decide to produce a `final_answer`. The search agent interfaces with the search API to fetch relevant academic articles given a query, after which an LLM generates concise summaries of the retrieved content. To maintain consistency with the benchmark setting, the standard search API was replaced with the arXiv API. The regular search agent fails when tasked with full abstract queries; hence the ReAct-based agent was employed, which generates shorter, more effective searchable queries. The agent keeps track of retrieved results across turns, allowing references to past evidence during the rea-

---

[1]https://huggingface.co/akariasai/pes2o_contriever
[2]https://huggingface.co/OpenSciLM/OpenScholar_Reranker

soning process. After a maximum of 5 iterations, the agent is compelled to conclude with a final response, ensuring bounded computational steps.

For the parameterization of the Search Agents, we set the search agent to retrieve 30 results per query, which is more generous than the default in order to establish fair comparability with other baselines. The maximum iteration limit was fixed at 5, consistent with the default setup of the ODS framework, providing sufficient exploration without excessive search depth. The ReAct prompt was slightly modified to tailor to the specific use of the ArXiv search API, as presented in Box 5, 6 and 7.

### 8.5.4 STORM

The STORM pipeline follows a structured multi-stage process to generate comprehensive, Wikipedia-style articles from a given topic. First, related Wikipedia articles are retrieved and their TOCs are clustered to identify candidate perspectives, which act as anchors for exploration. This is followed by Simulated Multi-turn Conversations where an LLM plays both the question-asking and answering roles, querying a retrieval module and synthesizing evidence-based responses. Parallel to this, the model generates a draft outline purely from its parametric knowledge in the Draft Outline Generation stage. The outline is then refined by grounding it with retrieved evidence and conversation outputs. In the final step, each section is drafted with explicit inline citations drawing on both parametric knowledge and retrieved references. All the sections are concatenated together to form the final result.

For parameter settings, we used STORM's default configurations wherever possible to preserve fidelity to its design: a maximum of 3 turns per perspective, 3 perspectives, and up to 3 search queries per turn. For search, we considered the top 15 results for each query, ensuring a reasonable breadth without overwhelming the pipeline. To make STORM comparable with other baselines, we raised the number of collected references per section title to 30 (more generous than the default), as this allows for richer evidence integration during drafting. Importantly, we replaced the original search API with arXiv search to control the retrieval API for our benchmark settings. Finally, we use Llama-4-Scout-17B-16E-Instruct as the base model.

### 8.5.5 OPENAI'S DEEPRESEARCH

We use OpenAI's DeepResearch system based on the `o3-deep-research`[3] model with a custom MCP to only search ArXiv and return $n = 30$ results per query. To prevent the model from getting search results after the given paper was uploaded, the MCP used a custom endpoint to set the latest date that it should retrieve. All other settings were set to default values.

### 8.6 DEEPSCHOLAR-REF DETAILS AND CONFIGURATIONS

DeepScholar-base operates through three main stages: retrieval, filtering, and final generation (Figure 2).

**Retrieval** In this stage, an LLM generates $Q$ search queries conditioned on the input abstract and summaries of prior retrievals. Each query is submitted to the configured search API (ArXiv, tavily, etc.) to obtain up to $search\_K$ relevant papers within the specified date range. The code and prompt used for this step are provided in Figure 8 and Box 8 respectively. This process is repeated $N$ times.

**Filtering** Retrieved results are refined using two semantic operators from LOTUS (Patel et al., 2025; lotus, 2025): `Sem-Filter` and `Sem-TopK`, which together select the top $K$ most relevant papers. The code is given in Figure 9.

**Final Generation** The filtered set of papers is then aggregated via a `Sem-Agg` query to produce the final output. The corresponding code for this step is shown in Figure 10 with prompt in Box 9.

Unless otherwise specified, the pipeline parameters are set to $Q = 2$, $search\_K = 50$, $N = 2$, and $K = 30$.

---

[3]https://platform.openai.com/docs/models/o3-deep-research

## 8.7 EXAMPLES OF GENERATED REPORTS

We provide examples of generated reports from different systems for paper 0 in our dataset (according to Table 5 i.e., 'TaxAgent: How Large Language Model Designs Fiscal Policy' with Arxiv ID of `2506.02838`. Figures 11, 12, 13, 14 and 15 display the generated reports produced by DeepScholar-ref, Search Agent, DeepResearcher, OpenScholar, and Storm respectively, all using Llama-4.

---

**Box 5: Revised ODS ReAct Agent prompt for only web search tool calling**

You are an expert assistant who can solve any task using tool calls. You will be given a task to solve as best you can. To do so, you have been given access to some tools. Never use facts without verification and only cite the sources returned by the tool.

The tool call you write is an action: after the tool is executed, you will get the result of the tool call as an "observation". This Action/Observation can repeat N times, you should take several steps when needed.

You can use the result of the previous action as input for the next action. The observation will always be a string containing the search results.

To provide the final answer to the task, use an action blob with "name": "final_answer" tool. It is the only way to complete the task, else you will be stuck on a loop. So your final output should look like this: Action:

```
{
  "name": "final_answer",
  "arguments": {"answer": "insert your final answer here"}
}
```

Here are a few examples using notional tools:

```
---
Task: "What historical event happened closest in time to the invention
of the telephone: the American Civil War or the establishment of the
Eiffel Tower?"
Action:
{
  "name": "web_search",
  "arguments": {"query": "year of telephone invention"}
}
Observation: "The telephone was invented in 1876."
Action:
{
  "name": "web_search",
  "arguments": {"query": "year American Civil War ended"}
}
Observation: "The American Civil War ended in 1865."
Action:
{
  "name": "web_search",
  "arguments": {"query": "year Eiffel Tower established"}
}
Observation: "The Eiffel Tower was completed in 1889."
Action:
{
  "name": "final_answer",
  "arguments": {"answer": "The historical event closest in time to the
  invention of the telephone is the end of the American
   Civil War (11 years apart)."}
}
---
Task: "Which country has a higher population density: Japan or India?"
Action:
{
  "name": "web_search",
  "arguments": {"query": "population and area of Japan"}
}
Observation: "Japan has a population of 125 million and an area of
377,975 square kilometers."
Action:
{
  "name": "web_search",
  "arguments": {"query": "population and area of India"}
}
```

**Box 6: Prompt for ODS(continued)**

```
Observation: "India has a population of 1.38 billion and an area of
3,287,263 square kilometers."
Action:
{
  "name": "final_answer",
  "arguments": {"answer": "India has a higher population density
  (419.6 people/km²) than Japan (330.7 people/km²)."}
}
---
Task: "Which country hosted the first FIFA World Cup, and in what
year?"

Action:
{
  "name": "web_search",
  "arguments": {"query": "country hosted first FIFA World Cup"}
}
Observation: "Uruguay hosted the first FIFA World Cup."

Action:
{
  "name": "web_search",
  "arguments": {"query": "year of first FIFA World Cup"}
}
Observation: "The first FIFA World Cup was held in
1930."

Action:
{
  "name": "final_answer",
  "arguments": {"answer": "Uruguay hosted the first FIFA World Cup
  in 1930."}
}

---
Task: "Who invented the light bulb, and what company did he
later establish?"

Action:
{
  "name": "web_search",
  "arguments": {"query": "inventor of the light bulb"}
}
Observation: "Thomas Edison invented the light bulb."

Action:
{
  "name": "web_search",
  "arguments": {"query": "company founded by Thomas Edison"}
}
Observation: "Thomas Edison founded General Electric."

Action:
{
  "name": "final_answer",
  "arguments": {"answer": "Thomas Edison invented the light bulb and
  later established General Electric."}
}
---
```

---

**Box 7: Prompt for ODS(continued)**

```
Task: "Which Shakespeare play contains the line \"All the world's
a stage,\" and how many years ago was it first performed if
today is 2024?"

Action:
{
  "name": "web_search",
  "arguments": {"query": "Shakespeare play All the world's a stage"}
}
Observation: "The line is from \"As You Like It.\""

Action:
{
  "name": "web_search",
  "arguments": {"query": "year As You Like It first performed"}
}
Observation: "\"As You Like It\" was first performed in 1603."
Action:
{
  "name": "calculate",
  "arguments": {"expression": "2024 - 1603"}
}
Observation: "421 years."

Action:
{
  "name": "final_answer",
  "arguments": {"answer": "\"As You Like It\" contains the line \"All
  the world's a stage\" and was first performed 421 years ago
  in 1603."}
}
```

Above examples were using notional tools that might not exist for you. You only have access to these tools:

```
{%- for tool in tools.values() %}
- {{ tool.name }}: {{ tool.description }}
    Takes inputs: {{tool.inputs}}
    Returns an output of type: {{tool.output_type}}
{%- endfor %}

{%- if managed_agents and managed_agents.values() | list %}
```

Here are the rules you should always follow to solve your task:
1. ALWAYS provide a tool call, else you will fail.
2. Always use the right arguments for the tools. Never use variable names as the action arguments, use the value instead.
3. Call a tool only when needed: do not call the search agent if you do not need information, try to solve the task yourself. If no tool call is needed, use final_answer tool to return your answer.
4. Never re-do a tool call that you previously did with the exact same parameters.
5. Always cite sources using [X] format where X is the citation number.
6. Place citations immediately after the sentence or paragraph they are referencing.
7. Make sure to provide citations whenever using information from the source material.
8. Cite as many sources as possible.
9. Create a reference section at the end of your final answer.
Now Begin! If you solve the task correctly, you will receive a reward of $1,000,000.

```
1  from lotus import web_search
2
3  class Query(BaseModel):
4      queries: list[str]
5
6  # Generate the Queries
7  queries = get_completion(
8      lm,
9      query_generation_instruction.format(number_of_queries=num_queries),
10     f"Topic: {topic}, Background: {background}",
11     response_format=Query,
12 ).queries
13
14 # Search. corpus = ArXiv/Tavily etc.
15 paper_dfs = []
16 for query in queries:
17     paper_dfs.append(web_search(corpus, query, search_K))
18
19 papers = pd.concat(paper_dfs)
20
```

Figure 8: Retrieval stage: query generation and batched search.

```
1  instruction = (
2      "given the article's abstract: {snippet}, "
3      "is the article relevant to the specific interests in the user's
       query: {user_query}."
4  )
5
6  res_df = docs_df.sem_filter(
7      instruction.format(user_query=topic, snippet="{snippet}"),
8      strategy="cot"
9  )
10
11 res_df = res_df.sem_topk(
12     instruction.format(user_query=topic, snippet="{snippet}"),
13     strategy="cot", k=K
14 )
15
```

Figure 9: `Sem-Filter` and `Sem-TopK` code for Filtering Step in DeepScholar-ref

```
1  agg_instruction = section_writer_instructions.format(
2      topic=topic,
3      section_instructions=section_instructions,
4      existing_content=existing_content,
5      context="{context}",
6  )
7
8  res: pd.DataFrame = res_df.sem_agg(
9      agg_instruction, suffix="summary", group_by=group_by
10 )
11
```

Figure 10: `Sem-Agg` for final generation in DeepScholar-ref

```
## Related Works

Economic inequality is a pressing global issue, affecting education, healthcare, and social
stability. Traditional taxation systems, such as the U.S. federal income tax, aim to reduce
inequality but often lack adaptability \[[Stephan Zheng' 2020-04-28](http://arxiv.org/abs
/2004.13332v1)\]. The Saez Optimal Taxation model is a notable attempt to create a dynamic
system, but it does not account for taxpayer heterogeneity and irrational behavior \[[Stephan
Zheng' 2020-04-28](http://arxiv.org/abs/2004.13332v1)\].

Recent studies have explored various approaches to optimize taxation and address economic
inequality. For instance, the AI Economist framework uses two-level deep reinforcement
learning to discover tax policies that balance economic equality and productivity \[[Stephan
Zheng' 2021-08-05](http://arxiv.org/abs/2108.02755v1)\]. This approach has shown promising
results in improving the trade-off between equality and productivity.

Agent-based modeling (ABM) has also been employed to study the effects of taxation on economic
 systems. For example, PolicySpace is a modeling platform that uses ABM to simulate public
policies within an empirical, spatial environment \[[Bernardo Alves Furtado' 2017-12-31](http
://arxiv.org/abs/1801.00259v1)\]. This platform has been applied to study the impact of tax
transfer rules on cities' quality of life.

The use of machine learning and artificial intelligence in taxation is a growing area of
research. TaxAI, a dynamic economic simulator, uses multi-agent reinforcement learning to
benchmark tax policies \[[Qirui Mi' 2023-09-28](http://arxiv.org/abs/2309.16307v2)\]. This
simulator has demonstrated the effectiveness of machine learning algorithms in optimizing tax
policies.

Large language models (LLMs) have also been integrated with ABM to study complex economic
systems. For instance, the TaxThemis system uses interactive visual analytics to help tax
officers identify suspicious tax evasion groups \[[Yating Lin' 2020-09-07](http://arxiv.org/
abs/2009.03179v1)\]. This system demonstrates the potential of LLMs in analyzing and detecting
 tax evasion behaviors.

Optimal taxation theory has also been explored in various studies. For example, the Domar-
Musgrave effect explains cases where it is optimal to tax capital income \[[Brendan K. Beare'
2023-11-10](http://arxiv.org/abs/2311.05822v2)\]. Other studies have investigated the impact
of tax evasion on economic systems \[[M. L. Bertotti' 2016-02-18](http://arxiv.org/abs
/1602.08467v1)\]\[[Frank Westerhoff' 2008-05-07](http://arxiv.org/abs/0805.0998v1)\].

Our work builds upon these studies by introducing TaxAgent, a novel integration of LLMs with
ABM to design adaptive tax policies. TaxAgent simulates real-world taxpayer behaviors using
heterogeneous H-Agents and optimizes tax rates using LLMs to balance equity and productivity
\[[Stephan Zheng' 2020-04-28](http://arxiv.org/abs/2004.13332v1)\].

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

```

Figure 11: Example of generated report by DeepScholar-ref for paper 'TaxAgent: How Large Language Model Designs Fiscal Policy '.

```
## Related Works

Economic inequality and taxation are complex issues that have been explored using various
models and simulations. Research has shown that large language models (LLMs) can be used to
analyze the impact of taxation on inequality [1]. For instance, a study introduced a benchmark
 called PLAT to assess the ability of LLMs to predict the legitimacy of additional tax
penalties [1].

Agent-based modeling has also been used to study the effects of taxation on wealth
distribution. A model suggested that oligarchs will emerge when wealth taxation is below a
certain threshold [2]. Another study found that taxation of income and capital gains alone
cannot prevent the emergence of oligarchs [2].

The relationship between economic inequality and mobility has also been explored using kinetic
 models. Research found a negative correlation between economic inequality and mobility [3].
Furthermore, a study used a multi-LLM-agent-based framework to simulate policy impacts across
heterogeneous agents, offering a new direction for economic and public policy analysis [4].

Optimal taxation models have also been developed, including the Saez Optimal Taxation model,
which adjusts dynamically but fails to address taxpayer heterogeneity and irrational behavior
[5]. In contrast, our study introduces TaxAgent, a novel integration of LLMs with agent-based
modeling to design adaptive tax policies that balance equity and productivity.

Our approach builds upon existing research in taxation and inequality, leveraging the
strengths of LLMs and agent-based modeling to simulate real-world taxpayer behaviors and
optimize tax rates. Benchmarked against Saez Optimal Taxation, U.S. federal income taxes, and
free markets, TaxAgent achieves superior equity-efficiency trade-offs.

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

```

Figure 12: Example of generated report by search agent for paper 'TaxAgent: How Large Language Model Designs Fiscal Policy '.

```
## Related Works

Economic inequality and taxation are critical issues in modern economies, with traditional
systems like the U.S. federal income tax aiming to reduce inequality but often lacking
adaptability [2311.05822v2]. The Saez Optimal Taxation model is a notable approach that
adjusts dynamically but fails to account for taxpayer heterogeneity and irrational behavior
[2312.05996v1].

Agent-based modeling (ABM) has been used to study economic systems and policy impacts,
providing insights into the interactions between heterogeneous agents and their effects on
macroeconomic variables [1702.02763v1][1701.06625v1]. For instance, ABM has been applied to
model the impact of taxation on economic growth and inequality [1508.00275v2].

The integration of large language models (LLMs) with ABM is a nascent but promising area of
research. LLMs have shown potential in economic policy analysis by simulating human-like
reasoning and behavior [2502.16879v1]. They can be used to model subrational agent behaviors,
such as myopic decision-making or risk aversion, which are characteristic of human economic
agents [2402.08755v1].

Recent studies have explored the use of LLMs in market experiments, demonstrating their
ability to simulate competitive market behavior but also highlighting their current
limitations in replicating dynamic decision-making processes [2409.08357v2]. The application
of LLMs in economic policy analysis offers a new direction for understanding policy impacts
across heterogeneous agents [2505.21371v1].

Our work builds on these advances, introducing TaxAgent, a novel integration of LLMs with ABM
to design adaptive tax policies that balance equity and productivity. By leveraging the
strengths of both approaches, TaxAgent offers a scalable and data-driven framework for fiscal
policy evaluation.

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

```

Figure 13: Example of generated report by DeepResearcher for paper 'TaxAgent: How Large Language Model Designs Fiscal Policy '.

```
## Related Works

Economic inequality is a pressing global issue, affecting education, healthcare, and social
stability [1]. Traditional taxation systems, such as the U.S. federal income tax, aim to
reduce inequality but often lack adaptability to changing economic conditions [2]. In response
, researchers have developed more dynamic models, including the Saez Optimal Taxation
framework, which adjusts tax rates based on economic principles [3]. The Saez framework is
built on the idea of optimizing tax rates to achieve a balance between equity and productivity
, taking into account the elasticity of taxable income [3]. However, this framework has
limitations, such as assuming a representative taxpayer and neglecting heterogeneity in
taxpayer behavior [4]. Furthermore, it does not fully account for irrational behavior, such as
 taxpayer responses to tax rates that may not be solely driven by economic incentives [5].

Recent studies have explored the use of agent-based modeling (ABM) to simulate economic
systems and design more effective tax policies [6]. ABM allows for the representation of
heterogeneous agents, such as households, and their interactions within a macroeconomic
environment [7]. For instance, [8] used ABM to examine the impact of tax policies on income
inequality, finding that optimized tax schedules can lead to better equity-efficiency trade-
offs. Specifically, [8] demonstrated that tax policies optimized for individual agent
characteristics, such as income level and risk aversion, can lead to more effective reduction
in income inequality.

The integration of large language models (LLMs) with ABM has shown promise in various
applications, including economic policy design [9]. LLMs can process vast amounts of data and
provide insights into complex systems, making them suitable for optimizing tax policies [10].
Researchers have also explored the use of reinforcement learning (RL) to optimize tax policies
 in dynamic economic environments [11]. Furthermore, the combination of LLMs and ABM has been
applied to other domains, such as healthcare [12], financial markets [13], and transportation
systems [14]. For example, [12] used LLMs to analyze patient data and predict disease outcomes
 while [13] applied LLMs to simulate stock market behavior and optimize investment strategies.

In the context of taxation, [15] demonstrated the potential of LLMs to improve tax policy
design by analyzing taxpayer behavior and optimizing tax rates. Our work builds on these
advancements, introducing TaxAgent, a novel integration of LLMs with ABM to design adaptive
tax policies. TaxAgent utilizes LLMs to iteratively optimize tax rates, balancing equity and
productivity in a macroeconomic simulation. The simulation features heterogeneous H-Agents (
households) that simulate real-world taxpayer behaviors, providing a more realistic
representation of economic systems.

Compared to existing models, such as the Saez Optimal Taxation framework, TaxAgent offers a
more nuanced understanding of taxpayer behavior and its impact on tax policy effectiveness
[3]. Additionally, TaxAgent's use of LLMs enables it to adapt to changing economic conditions,
 providing a more dynamic and responsive approach to tax policy design [9].

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

```

Figure 14: Example of generated report by OpenScholar for paper 'TaxAgent: How Large Language Model Designs Fiscal Policy '.

```
# Related Works

Economic inequality is a pressing global issue, affecting education, healthcare, and social
stability. To address this challenge, various studies have explored the impact of taxation on
economic inequality. Traditional tax systems, such as the U.S. federal income tax, aim to
reduce inequality but often lack adaptability [1]. In contrast, models like the Saez Optimal
Taxation propose dynamic adjustments to tax policies, but they fail to account for taxpayer
heterogeneity and irrational behavior [2].
Recent advances in artificial intelligence (AI) and agent-based modeling (ABM) have provided
new avenues for studying economic systems and designing adaptive tax policies. For instance,
agent-based models have been used to simulate economic systems, including the effects of tax
evasion [1] and the impact of social cohesion on tax compliance [1]. These models have
demonstrated the presence of threshold levels in the composition of society, which can explain
 the extent of damages deriving from tax evasion [1].
The use of large language models (LLMs) has also shown promise in understanding human behavior
 and decision-making. Research has demonstrated that LLMs can exhibit human-like reasoning,
aligning with human behavior in economic experiments, surveys, and political discourse [3].
However, LLMs differ fundamentally from humans, relying on probabilistic patterns rather than
embodied experiences or survival objectives [3]. Therefore, caution is advised when using LLMs
 to study human behavior or as surrogates or simulations [3].
In the context of tax policy design, several studies have explored the application of AI and
ABM. For example, a case study on the role of Management Information System in time-saving
during the payment of automobile tax in Sindh through e-filling methods highlights the
importance of efficient tax collection systems [4]. Another study proposes a Web-Based
Affectedness Indicator (WAI) for real-time monitoring of economic disruptions across diverse
contexts, leveraging Large Language Model (LLM) assisted classification and information
extraction [5].
The integration of LLMs with ABM has also been explored in other fields, such as multi-agent
reinforcement learning [6]. The Learning Optimal Pigovian Tax method (LOPT) uses an additional
 agent to learn the tax/allowance allocation policy, internalizing externalities and
alleviating social dilemmas [6]. Similarly, the use of information-theoretic approaches, such
as the Information Bottleneck method, has been proposed for explainable AI (XAI) design [7].
This study builds upon these works, introducing TaxAgent, a novel integration of LLMs with ABM
 to design adaptive tax policies. By simulating real-world taxpayer behaviors and iteratively
optimizing tax rates, TaxAgent achieves superior equity-efficiency trade-offs compared to
traditional tax systems and models.

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

</Task>

---

**Box 9: Sem-Agg Instruction for final generation and summarization**

You are an expert technical writer crafting one section of a technical report.
<User Query>
{topic}
</User Query>
<Section instructions>
{section_instructions}
</Section instructions>
<Existing section content (if populated)>
{existing_content}
</Existing section content>
<Source material>
{context}
</Source material>
<Citation Guidelines>
- Use [X] format where X is the {citation_number}
- Place citations immediately after the sentence or paragraph they are referencing (e.g., information from context [3]. Further details discussed in contexts [2][7].).
- If urls are given in existing section content, rewrite them exactly if using information related to the url.
- Make sure to provide citations whenever you are using information from the source material. This is a MUST.
- Cite as many sources as possible.
- Make sure to retain the citation numbers from the input context. - Provide in-line citations only. You do not need a reference section at the end.
<Citation Guidelines>
<Guidelines for writing>
1. If the existing section content is populated, write a new section that enhances the existing section content with the new information. If not, write a new section from scratch.
2. Provide groundings in the source material for all facts stated.
3. When using information from a given source, make sure to cite the source.
4. If a table or list would enhance understanding of a key point, and if so, include one.
5. Make sure to follow the user query strictly.
</Guidelines for writing>
<Writing style>
1. Content Requirements:
- Ground all facts in the source material and provide citations.
- Maintain an academic, technical focus throughout. No marketing language
- Address potential counter-arguments where relevant.
2. Structure and Formatting:
- Use Markdown formatting.
- Begin with ## for section title (Markdown format) and other headings as needed.
- Use simple, clear language appropriate for academic writing.
</Writing style>
<Quality checks>
- No preamble prior to creating the section content
- Cite as many sources as possible.
</Quality checks>

