# OpenReview forum: "DeepScholarBench: A Live Benchmark and Automated Evaluation for Generative Research Synthesis"
_ICLR.cc/2026/Conference — Submitted to ICLR 2026_

### Official Review · Reviewer_ozAV · 2025-10-23

**Soundness:** 2
**Presentation:** 3
**Contribution:** 2
**Rating:** 4
**Confidence:** 3

**Summary:**

This paper introduces a new benchmark and evaluation framework for assessing related work writing in research papers. The benchmark includes 63 recently published papers spanning 18 distinct domains. The evaluation framework measures performance on seven fine-grained metrics covering knowledge synthesis, retrieval quality, and verifiability. Experimental results reveal the challenging nature of this task, as no system achieves a geometric mean exceeding 31% across all metrics.

**Strengths:**

1. This is the first paper to evaluate the related-work writing capabilities of the Deep Research system.

2. The proposed evaluation framework provides comprehensive coverage of key aspects.

3. The paper is clearly written and easy to follow.

**Weaknesses:**

1. In lines 174–175, the notation description is somewhat confusing. I suggest placing each symbol immediately after its corresponding concept (e.g., S should directly follow “a set of relevant sources” rather than being separated by a comma). Additionally, consider separating different notations and their descriptions with semicolons to avoid reader confusion.

2. For quotation marks, please use ``` `` ``` and ``` '' ``` to represent the left and right quotation marks, respectively.

3. If an error analysis were included here—identifying the causes and proportions of the error cases—it would provide valuable insights for future work.

**Questions:**

No

---

> ### Author Response · Authors · 2025-11-25
> **Response to Reviewer ozAV**
>
> > W1. In lines 174–175, the notation description is somewhat confusing. I suggest placing each symbol immediately after its corresponding concept (e.g., S should directly follow “a set of relevant sources” rather than being separated by a comma). Additionally, consider separating different notations and their descriptions with semicolons to avoid reader confusion.
>
> Thank you for this suggestion and we have revised the manuscript with the update.
>
>
> > W2. For quotation marks, please use `` and '' to represent the left and right quotation marks, respectively.
>
> Thank you for this suggestion and we have updated the manuscript to improve our presentation.
>
>
> > W3. If an error analysis were included here—identifying the causes and proportions of the error cases—it would provide valuable insights for future work.
>
> We thank the reviewer for the suggestion, and we would like to clarify that in Section 6.2, we provide a detailed ablation study to understand causes of error and opportunities for improvement. Specifically our controlled experiment explores whether the performance limitations of existing systems lies in their retrieval capabilities or in their knowledge synthesis capabilities. To perform this analysis we benchmark two baselines, DeepScholar-ref (GPT-4.1, Claude) and DeepScholar-ref (Llama-4), using an oracle retriever setup. We find that this oracle setting nearly saturates retrieval quality, while knowledge synthesis remains low. This indicates that existing systems struggle both due to inefficiencies in retrieval, specifically with finding a holistic set of key reference (reflected by low reference coverage), and finding notable, influential references (reflected by low document importance), as well as inefficiencies in LLM capabilities to synthesize information and surface crucial facts (reflected by low nugget coverage). Notably, even with the oracle retrieval setting, DeepScholar-ref (GPT-4.1, Claude) obtains only .528 Nugget Coverage at best, indicating that existing LLMs struggle to effectively synthesize knowledge from dozens of documents even when the documents provide comprehensive factual coverage.

---

### Official Review · Reviewer_AnFp · 2025-10-30

**Soundness:** 3
**Presentation:** 2
**Contribution:** 2
**Rating:** 4
**Confidence:** 3

**Summary:**

This paper introduces DeepScholar-bench, a live benchmark for Generative Research Synthesis systems. The authors argue that existing benchmarks are ill-suited, as they are often static or focus on short-form answers. DeepScholar-bench addresses this by using recent ArXiv papers as a data source. The task is to generate a related work section given a paper's abstract. The framework evaluates systems on three dimensions: Knowledge Synthesis, Retrieval Quality, and Verifiability. The authors also provide an open-source baseline, DeepScholar-ref. An evaluation of systems shows the benchmark is far from saturated, with no system surpassing a 31% geometric mean score.

**Strengths:**

1. The paper tackles the highly significant problem of evaluating complex GRS systems. The core idea of using recent ArXiv papers as a live and continuously updating data source is novel and effectively mitigates data staleness and contamination.
2. The three-pillar evaluation framework is comprehensive and well-justified. The fine-grained metrics allow for a nuanced analysis of system capabilities
3. The thorough empirical study establishes the benchmark's difficulty, and the inclusion of the DeepScholar-ref baseline is a valuable contribution for future research

**Weaknesses:**

1. The use of human-written related work sections as exemplars is a key weakness. Human authors are constrained by page limits and their own limited knowledge. This means the exemplar may be incomplete, which unfairly penalizes an AI system that performs better retrieval or synthesis than the human author. This specifically impacts the validity of the Retrieval Quality and Verifiability metrics.
2. While the live aspect is valuable for GRS, the live aspect have explored in other domains [1,2,3], which should be acknowledged.
3. Using a paper's full abstract as the input query  does not seem to reflect a realistic user scenario for a GRS task. A user would more likely start with a research question or topic, not a completed abstract. This design choice may impact the generalizability of the results.

[1] HOH: A Dynamic Benchmark for Evaluating the Impact of Outdated Information on Retrieval-Augmented Generation
[2] PAT-Questions: A Self-Updating Benchmark for Present-Anchored Temporal Question-Answering
[3] MAC: A Live Benchmark for Multimodal Large Language Models in Scientific Understanding

**Questions:**

1. Given that human exemplars are imperfect, how does your evaluation framework account for a system that finds relevant, important papers missed by the human author? Would this not unfairly penalize the system on metrics like Reference Coverage?
2. Why was the full abstract chosen as the input query? How do you think system performance would differ if the prompt was a more realistic, shorter input, such as the paper's title or a one-sentence research question?

---

> ### Author Response · Authors · 2025-11-25
> **Response to Reviewer AnFp (1/2)**
>
> > W1. The use of human-written related work sections as exemplars is a key weakness. Human authors are constrained by page limits and their own limited knowledge. This means the exemplar may be incomplete, which unfairly penalizes an AI system that performs better retrieval or synthesis than the human author. This specifically impacts the validity of the Retrieval Quality and Verifiability metrics.
>
> > Q1. Given that human exemplars are imperfect, how does your evaluation framework account for a system that finds relevant, important papers missed by the human author? Would this not unfairly penalize the system on metrics like Reference Coverage?
>
> Thank you for raising this point, which is important for a fair and robust benchmark evaluation. We would like to clarify that among our metrics, Reference Coverage is the only one that relies on citation matches with references from the human exemplar. We carefully design this metric to avoid falsely penalizing systems, and have added a new human evaluation study, which validates the quality of this metric.  We provide a detailed explanation of our Retrieval and Verifiability metrics below and details from our new human evaluation study:
>
> - __Retrieval Quality__: We use three metrics designed to complement each other and provide a holistic evaluation:
>    - __Relevance Rate__ assigns a positive score to any retrieved source from the system generated report that is classified as relevant to the paper. These references need not be in the human exemplar.
>
>     - __Document Importance__ rewards systems which generate reports with highly-cited references. These references need not be in the human exemplar.
>    - __Reference Coverage__ provides a notion of recall over a set of key, important sources. We carefully design this metric by curating a set of “important references” from each human exemplar using an LLM-judge which marks each extracted reference as “important” or “not important”. To validate this metric we have provided a new human evaluation study.  We recruit 11 expert annotators, who are CS PhD or postdocs among 4 Universities’ CS departments and who are not authors of this paper. We collect over 100 blind annotations and report the confusion matrix of human-LLM agreement below for labeling references as “essential”. Overall the results demonstrate a low likelihood of falsely penalizing systems, given by the low False Positive Rate (9.8%). Further, we find a 65.9% agreement rate, demonstrating that our LLM-judge provides a strong heuristic for determining key references from the human exemplars without falsely penalizing systems that fail to cite non-essential references from the human exemplars. We also note that our data pipeline filters to select high-quality papers accepted for publication, helping us to maintain high-quality exemplars.
>
> | Actual / Predicted | Not Important  | Important |
> |--------------------|--------|-------|
> | Not Important      | 40.2 % | 9.8%  |
> | Important         | 24.2%  | 25.7% |
>
>
>
> - __Verifiability__: Both of our metrics, __Citation Precision__ and __Claim Coverage__, are computed independent of the human-written exemplar. Specifically, we process each generated report by extracting cited sources and sentence-level claims from the generated report and assess entailment relations. Our metrics assign positive scores to a report which fully supports its claims by its cited sources (i.e., claim coverage), and cites sources which support at least one claim in the generated report (i.e., citation precision). The cited sources and claims need not match the human exemplars.
>
> We have added details of our new human evaluation study to the manuscript (Section 6.3).

---

> > ### Author Response · Authors · 2025-11-25
> > **Response to Reviewer AnFp (2/2)**
> >
> > > W2. While the live aspect is valuable for GRS, the live aspect have explored in other domains [1,2,3], which should be acknowledged.
> > > [1] HOH: A Dynamic Benchmark for Evaluating the Impact of Outdated Information on Retrieval-Augmented Generation [2] PAT-Questions: A Self-Updating Benchmark for Present-Anchored Temporal Question-Answering [3] MAC: A Live Benchmark for Multimodal Large Language Models in Scientific Understanding
> >
> > We thank the reviewer for this point, and we agree that highlighting prior work on live benchmarks explored in other domains is valuable for a related work section. We have updated our manuscript to include this.
> >
> > > W3. Using a paper's full abstract as the input query does not seem to reflect a realistic user scenario for a GRS task. A user would more likely start with a research question or topic, not a completed abstract. This design choice may impact the generalizability of the results.
> >
> > > Q2. Why was the full abstract chosen as the input query? How do you think system performance would differ if the prompt was a more realistic, shorter input, such as the paper's title or a one-sentence research question?
> >
> > Thank you for your great feedback – this question raises a very insightful point warranting further study, which we have conducted and added to the manuscript (Appendix 8.3.3). While our study focuses on _one_ academic report-generation task, understanding the generalizability of our benchmark to alternative query formulations is valuable. As such, we have conducted an ablation to compare three formulations of the query:
> > - Using the abstract to describe the paper’s scope (abstract)
> > - Using a two-sentence summary of the paper (key idea)
> > - Using a research question (RQ)
> >
> > We generate the two-sentence summary and research question versions of the query using an LLM to convert the paper abstract to the respective alternatives. We then measure performance of 6 baselines on the full dataset for each query version. Below we have reported pearson correlation and p-values obtained using a permutation test.
> >
> > Overall, the results support the generalizability of our benchmark task and query formulation. We find high, significant correlation of each of our metrics between the abstract-based query and either alternative query-formulations. Our choice of directly using the abstract for the query allows us to build an efficient dataset pipeline, directly scraping from existing papers, while still providing generalizable results.
> >
> > | Metric                               | Org.  | Nug Cov.  | Rel. Rate  | Ref. Cov. | Doc. Imp. | Cite-P | Claim Cov. |
> > |--------------------------------------|-------|-------|-------|---------|---------|--------------------|----------------|
> > | Pearson corr (abstract,  key idea)   | 0.997 | 0.980 | 0.979 | 0.992   | 0.992   | 0.589              | 0.841          |
> > | P-value (abstract,  key idea)       | 0.012 | 0.006 | 0.003 | 0.007   | 0.053   | 0.112              | 0.014          |
> > | Pearson corr (abstract, RQ)         | 0.988 | 0.979 | 0.772 | 0.951   | 0.707   | 0.766              | 0.877          |
> > | P-value (abstract,  RQ)             | 0.014 | 0.003 | 0.010 | 0.004   | 0.079   | 0.046              | 0.017          |

---

> > > ### Comment · Reviewer_AnFp · 2025-11-26
> > >
> > > Thank you for your detailed response. Most of my concern have been addressed and I will raise rating to 6.  Good luck to you.

---

### Official Review · Reviewer_s3EJ · 2025-10-31

**Soundness:** 3
**Presentation:** 2
**Contribution:** 3
**Rating:** 4
**Confidence:** 3

**Summary:**

This paper presents DeepScholar-Bench, a live and automated benchmark designed to evaluate large-language-model systems that perform generative research synthesis—that is, retrieving, analyzing, and writing related-work sections with proper citations. The benchmark constructs monthly datasets from recent arXiv papers across 18 computer-science domains (63 papers in the June-2025 release), pairing each paper’s abstract with its human-written related-work section as an exemplar. The authors introduce a unified evaluation framework measuring three aspects: knowledge synthesis (organization and coverage), retrieval quality (relevance, coverage, and document importance), and verifiability (citation precision and claim coverage).
To provide a reproducible reference system, they also release DeepScholar-Ref, a modular open-source pipeline built atop the LOTUS semantic-operator framework that performs retrieval, filtering, ranking, and aggregation using LLMs. Experiments compare open-source, commercial, and search-agent systems, showing that current models reach only about 30% of human-level performance—highlighting the difficulty and future potential of reliable automated research synthesis.

**Strengths:**

The paper’s originality lies in formally defining generative research synthesis as a measurable task and introducing the first live benchmark for it. Rather than focusing on static QA or summarization datasets, the authors frame related-work generation as an end-to-end process involving retrieval, synthesis, and citation verification—an elegant combination of multiple research threads that were previously studied in isolation. The inclusion of verifiability metrics (Citation Precision and Claim Coverage) is particularly novel, addressing a key gap in evaluating factual grounding within long-form academic writing.
In terms of quality, the benchmark is thoughtfully engineered: papers are carefully filtered for publication status and clean structure, citations are resolved through both arXiv and OpenAlex, and evaluation criteria are explicitly defined and reproducible. The accompanying DeepScholar-Ref pipeline demonstrates solid engineering discipline, showing how an LLM can be operationalized into a modular retrieval-and-synthesis workflow.
The paper also excels in clarity. The data-collection process, metric design, and limitations are clearly presented with transparent assumptions and detailed ablations (e.g., retrieval-API comparison, oracle analysis). Figures and tables effectively communicate both methodology and findings, and the authors are candid about underestimations in human verifiability metrics.
Finally, the significance is high: DeepScholar-Bench provides a foundation for evaluating an emerging class of “research-assistant” systems. Its live, renewable design enables longitudinal comparison of academic-generation models and encourages more rigorous work on reliable citation and retrieval. The combination of benchmark, metrics, and open reference pipeline makes this paper a strong step toward standardizing the evaluation of LLM-based scientific writing systems.

**Weaknesses:**

## Field- and age-bias in Document Importance (DI)
The DI metric relies on median raw citation counts, which vary substantially by research field and publication year. Even with a median, the metric can still privilege older or citation-dense areas (e.g., NLP) and disadvantage emerging domains. Newly published papers inherently carry fewer citations regardless of relevance. The authors acknowledge this limitation; as a result, DI may misrepresent impact across domains and over time.

## Limited dataset scale and representativeness
The benchmark currently includes 63 papers across 18 CS subfields. While diverse—and each related-work section averages ~23 citations under strict filtering criteria—the small sample raises two concerns: (i) whether the dataset size ensures statistical robustness for metric/model comparisons, and (ii) whether specialized or cross-disciplinary research is underrepresented.
Aggregating multiple monthly snapshots is non-trivial due to contamination risk (if models have seen prior releases). Independent of contamination, domain coverage remains limited at present.

## Subjectivity in citation importance labeling
Reference Coverage assumes the exemplar’s “important” citations constitute the gold set, yet “importance” is implicitly shaped by the original author’s citation style and emphasis, which can introduce bias. Without additional annotation or inter-rater validation, can this metric truly reflect an objective notion of citation importance?

## Noise and underestimation in verifiability metrics
Citation Precision and Claim Coverage depend on LLM-based entailment, which can be noisy. For human exemplars, the absence of sentence-level evidence spans systematically underestimates verifiability (as the authors note). Given the benchmark’s automated setup, the current verifiability numbers should be interpreted with caution, especially when comparing humans to systems.

**Questions:**

## DI calibration (field & age effects)
Is it possible to report DI stratified by publication year and cs.* domain (or provide a simple normalization ablation) to show whether DI rankings change materially?

## Reliability & stability of results
What is the run-to-run variability for both generation and LLM-as-judge decisions, and which metrics are most vs. least stable given the 63-query size (e.g., confidence intervals or similar estimates)?

## Reliability of important citation labels
Since RC hinges on “important” vs. “non-important” citations in exemplars, what methods support the reliability of this labeling (e.g., rubric/prompt specification, inter-rater checks, or ensemble agreement)?

## Where is the main bottleneck? (oracle ablation)
Given the large gains under oracle retrieval, can you break down per-stage contributions (retrieval vs. filtering vs. ranking) to identify the dominant bottleneck?

---

> ### Author Response · Authors · 2025-11-25
> **Response to Reviewer s3EJ (1/4)**
>
> > W1. Field- and age-bias in Document Importance (DI)
> The DI metric relies on median raw citation counts, which vary substantially by research field and publication year. Even with a median, the metric can still privilege older or citation-dense areas (e.g., NLP) and disadvantage emerging domains. Newly published papers inherently carry fewer citations regardless of relevance. The authors acknowledge this limitation; as a result, DI may misrepresent impact across domains and over time.
>
> > Q1. DI calibration (field & age effects)
> Is it possible to report DI stratified by publication year and cs.* domain (or provide a simple normalization ablation) to show whether DI rankings change materially?
>
> We thank the reviewer for this clarification and we agree that providing a metric capable of assessing a variety of papers from different years and domains is important for our benchmark. We would like to clarify that we designed our Document Importance metric to be normalized to avoid this pitfall. Specifically, we compute the score by taking the ratio of the median number of citations per reference in the generated report to the median number of citations per reference in the human exemplar. For research domains that are less well-cited, the score’s denominator, computed from the human-exemplar to be lower, allowing us to avoid penalizing system generated reports with relatively low number of citations per reference. In our reported results, we take the average over all reports, allowing each paper, regardless of its domain, to carry equal weight.

---

> ### Author Response · Authors · 2025-11-25
> **Response to Reviewer s3EJ (2/4)**
>
> >W2. Limited dataset scale and representativeness
>  The benchmark currently includes 63 papers across 18 CS subfields. While diverse—and each related-work section averages ~23 citations under strict filtering criteria—the small sample raises two concerns: (i) whether the dataset size ensures statistical robustness for metric/model comparisons, and (ii) whether specialized or cross-disciplinary research is underrepresented. Aggregating multiple monthly snapshots is non-trivial due to contamination risk (if models have seen prior releases). Independent of contamination, domain coverage remains limited at present.
>
> Thanks you for this constructive feedback, and we agree that domain coverage and contamination are two crucial aspects of our benchmark. We have conducted new experiments using an expanded dataset, and we provide an explanation of the two key issues raised below.
>
> __Domain Coverage and Dataset Size__  We have addressed this by adding an additional study over an expanded dataset, DeepScholar-Nov-2025, demonstrating the generalizability of our benchmark. This dataset includes 200 queries scraped from a diverse set of over 75 arxiv domains, which include both papers from ArXiv’s CS, Biology, Physics, Economics and Finance servers, including ones from specialized or cross-domain categories. We report the performance of high-performing baselines with both open-source and closed-source models below and have added these results to our manuscript (Appendix 8.3.2). Additionally, we have added markers to our main results in Table 2 over DeepScholar-June-2025 to mark statistical significance when comparing the best-performing baseline to the others for each metric.
>
> Table 8: Performance of selected systems on DeepScholar-Nov-2025 (200 queries across 75 arXiv subject areas)
> | Model                          | Org   | Nug   | Rel Rate | Ref Cov | Doc Imp | Cite P | Claim Cov |
> |--------------------------------|-------|-------|----------|---------|---------|--------|-----------|
> | DeepScholar (Llama-4-Scout)    | 0.120 | 0.358 | 0.395    | 0.072   | 0.082   | 0.178  | 0.581     |
> | DeepScholar (GPT4.1 + o3)      | 0.578 | 0.479 | 0.568    | 0.087   | 0.056   | 0.563  | 0.578     |
> | Search AI (Llama-4-Scout)      | 0.108 | 0.252 | 0.179    | 0.034   | 0.079   | 0.189  | 0.314     |
> | Search AI (o3)                 | 0.608 | 0.480 | 0.623    | 0.078   | 0.034   | 0.475  | 0.452     |
>
>
>
> We highlight that our work proposes an automated dataset-curation pipeline, rather than a fixed dataset, and additional domains can be easily configured by our pipeline for practitioners to run. Our core analysis and main results in the paper remain on the DeepScholar-June-2025, which focuses on computer science papers. This choice was deliberate in order to allows us to perform systematic validation of our benchmark metrics and design with expert annotators consisting of CS PhDs and postdocs.
>
> Notably, we observe patterns from our evaluation on DeepScholar-Nov-2025 that are consistent with our main results on DeepScholar-June-2025. For both DeepScholar-ref and the Search Agent baseline, the o3-based variants substantially outperform their Llama-4-Scout counterparts on Organization, Nugget Coverage, Relevance Rate, Citation Precision, and Claim Coverage, while Reference Coverage and Document Importance remain broadly low across baselines. These trends mirror the relative ranking and qualitative gaps seen in our main benchmark slice analyzed in the paper, suggesting that our conclusions from DeepScholar-June-2025 generalize beyond queries related to Computer Science.
>
> __Contamination__ We clarify that our live benchmark’s leaderboard will avoid contamination by re-running baselines on _the most recent dataset only_ rather than aggregating datasets taken from different time snapshots.

---

> > ### Author Response · Authors · 2025-11-25
> > **Response to Reviewer s3EJ (3/4)**
> >
> > > W3. Subjectivity in citation importance labeling
> > Reference Coverage assumes the exemplar’s “important” citations constitute the gold set, yet “importance” is implicitly shaped by the original author’s citation style and emphasis, which can introduce bias. Without additional annotation or inter-rater validation, can this metric truly reflect an objective notion of citation importance?
> >
> > > Q3. Reliability of important citation labels
> > Since RC hinges on “important” vs. “non-important” citations in exemplars, what methods support the reliability of this labeling (e.g., rubric/prompt specification, inter-rater checks, or ensemble agreement)?
> >
> > Thank you for raising this point, which is important to providing a robust evaluation of retrieval quality. We have added an additional human ablation study to provide further analysis, which validates the quality of our Reference Coverage metric. Specifically, we recruit 11 expert annotators, who are CS PhD or postdocs among 4 Universities’ CS departments and who are not authors of this paper. We collect over 100 blind annotations for this metric and report the confusion matrix of human-LLM agreement below for labeling references as “essential”. Overall the results demonstrate a low likelihood of falsely penalizing systems, given by the low False Positive Rate (9.8%). Furthermore, the results demonstrate our LLM-judge provides a strong heuristic for determining key references from the human exemplars with a 65.9% agreement rate. We have added this study to our manuscript (Section 6.3). Additionally, we provide all prompts used for our LLM judges in Appendix Section 8.4, and we implement each without calibration. Our prompt for the LLM-judge used to judge the reference importance is shown in the Appendix Figure 5.
> >
> > | Actual / Predicted | Not Important  | Important  |
> > |--------------------|--------|-------|
> > | Not Important      | 40.2 % | 9.8%  |
> > | Important         | 24.2%  | 25.7% |
> >
> > > W4. Noise and underestimation in verifiability metrics
> > Citation Precision and Claim Coverage depend on LLM-based entailment, which can be noisy. For human exemplars, the absence of sentence-level evidence spans systematically underestimates verifiability (as the authors note). Given the benchmark’s automated setup, the current verifiability numbers should be interpreted with caution, especially when comparing humans to systems.
> >
> > Thank you for this feedback on our presentation, and we agree that our current verifiability results must be interpreted with caution. To aid the reader we have added an additional manual validation study, computing the citation precision and citation coverage from a sample of 20 papers. We obtain scores of 90% Citation Precision and 85% Citation Coverage of the human exemplars from our study. We have updated the manuscript to include these numbers in Table 2.

---

> > > ### Author Response · Authors · 2025-11-25
> > > **Response to Reviewer s3EJ (4/4)**
> > >
> > > > Q2. Reliability & stability of results
> > > What is the run-to-run variability for both generation and LLM-as-judge decisions, and which metrics are most vs. least stable given the 63-query size (e.g., confidence intervals or similar estimates)?
> > >
> > > Thank you for the question. We clarify that in Figure 1 we provide a detailed breakdown of each metric’s performance range among all baselines for our 63-query data size. We observe that the verifiability metrics, Citation Precision and Claim Coverage exhibit the largest performance range. To validate the correctness of the LLM-judges used in our metrics, we perform an manual validation shown in the Appendix Table 7 from 400 expert annotations. These results show an 80% agreement relation for the entailment judges used in our validation. Furthermore, our qualitative analysis based on examples, as shown in Appendix Section 8.5 and quantitative analysis from Appendix Table 5 confirms that our measured baselines exhibit a wide range of behaviors in terms of the number of inline citations and frequency of citations, which explains the high variance in Citation Precision and Claim Coverage scores among baselines.
> > >
> > >
> > > > Q4. Where is the main bottleneck? (oracle ablation)
> > > Given the large gains under oracle retrieval, can you break down per-stage contributions (retrieval vs. filtering vs. ranking) to identify the dominant bottleneck?
> > >
> > > Thank you for this question raising an clarification for our ablation study. We would like to clarify that our oracle method _skips_ the retrieval, filtering and ranking steps, directly feeding the LLM the oracle corpus. The performance gain seen by the oracle retrieval method is instead attributable to the LLM citing and synthesizing the provided sources, leading to saturated retrieval quality as well as higher nugget coverage.

---

### Official Review · Reviewer_ijFd · 2025-11-01

**Soundness:** 4
**Presentation:** 4
**Contribution:** 4
**Rating:** 8
**Confidence:** 5

**Summary:**

DeepScholarBench introduces a new live benchmark and automated evaluation framework for generative research synthesis systems. The benchmark constructs realistic tasks by scraping recent, peer-reviewed ArXiv papers and using their related work sections as high-quality exemplars. Each task provides a paper’s title/abstract as input and requires an AI system to retrieve relevant literature, synthesize a long-form related work section, and cite sources. The authors contribute an automated data pipeline that continually updates the dataset (e.g. monthly) to avoid staleness. They also develop a holistic evaluation measuring knowledge synthesis, retrieval quality, and verifiability via seven fine-grained metrics, which show strong agreement with human judgments. Additionally, the paper provides DeepScholar-ref, an open-source reference pipeline built on the LOTUS framework as a baseline solution. Extensive experiments evaluate 14 baseline systems – including open-source methods (e.g. STORM, OpenScholar), search-agent pipelines with LLMs (GPT-4.1, Claude, Gemini, etc.), and OpenAI’s DeepResearch – on the benchmark. The results show significant room for improvement: no system exceeds ~31% (geometric mean) across the metrics, highlighting the difficulty of true research synthesis and the value of DeepScholarBench for driving progress.

**Strengths:**

The paper targets an important problem: how to robustly evaluate AI systems on complex research synthesis tasksbeyond simple Q&A. Its originality lies in creating a live, continuously evolving benchmark – a forward-looking design that avoids the pitfalls of static datasets becoming outdated. The methodology is high-quality: the authors build a careful data pipeline to source realistic tasks from recent accepted papers, enforcing quality controls (conference acceptance, proper references) to ensure the evaluation set is credible. The evaluation is holistic and well-motivated: instead of a single score, it breaks performance down into understandable dimensions (writing coherence, coverage of essential content, retrieval relevance, source importance, citation correctness), providing a nuanced view of system capabilities. Automating these metrics with LLM-based judges and entailment checks is innovative and enables scalable, repeatable evaluation – notably, the authors verify these automated metrics against human judgments, lending credibility to their approach. The clarity of the exposition is another strength: the paper is organized with the reader in mind, offering conceptual figures (e.g. illustrating the pipeline and evaluation framework) and concrete examples. Lastly, the significance is clear – DeepScholarBench fills a crucial gap by providing a benchmark that current state-of-the-art models struggle on (none exceeding ~39% on key metrics), thereby establishing a challenging platform that will likely spur significant research and improvements in generative research assistants.

**Weaknesses:**

1. **LLM-based evaluation bias**: The reliance on LLM judges (e.g., for Organization & Coherency) may favor formulaic writing over genuine quality. Adding human evaluation or refining prompts could help validate that metrics reflect true scholarly standards.
2. **Reference coverage limitations**: The metric penalizes systems for citing valid alternative papers not in the human reference list, rather than evaluating whether different sources cover the same content. Consider assessing content coverage instead of exact reference matching.
3. **Narrow scope**: The dataset covers only 63 CS queries from one quarter. Expanding to other domains (biomedical, social sciences) and creating a fixed benchmark subset alongside monthly updates would improve generalizability and enable longitudinal comparisons.
4. **Weak citation detection**: The baseline (DeepScholar-ref) scores near 0% on Document Importance, missing high-impact papers. Systems should incorporate citation metrics or seminal paper detection to identify influential references.

**Questions:**

1. Reference Coverage Metric: How are “important” references determined in the human exemplar? Is this labeling done manually or via an automated heuristic/LLM? Clarifying this process (and whether alternative relevant sources are somehow acknowledged) would help us interpret the reference coverage scores.
2. Generality: The dataset currently draws from computer science papers. Do the authors plan to extend DeepScholarBench to other research domains (e.g. medicine, physics), or is the focus primarily on CS/AI literature? Expanding domain coverage could increase the benchmark’s impact.
3. Live Benchmark Dynamics: With monthly updates, how will the authors ensure fair comparison over time? For example, if a new model is evaluated on a future set of queries, will there be a way to compare its performance to the scores reported in this paper (on the June 2025 set)? It would be useful to know if a static test subset or leaderboard versioning is planned to handle the evolving data.
4. Baseline Pipeline Improvements: DeepScholar-ref performs well, but notably struggles with retrieving highly-cited documents (Document Importance ≈0.01). What are the authors’ thoughts on augmenting the pipeline with an “importance awareness” (e.g. factoring in citation counts or influence of sources during retrieval)? This could potentially boost that metric – have such improvements been considered for future versions of the reference pipeline?

---

> ### Author Response · Authors · 2025-11-25
> **Response to Reviewer ijFd (1/3)**
>
> > W1. LLM-based evaluation bias: The reliance on LLM judges (e.g., for Organization & Coherency) may favor formulaic writing over genuine quality. Adding human evaluation or refining prompts could help validate that metrics reflect true scholarly standards.
>
> Thank you for raising these important concerns regarding our use of LLM-judges and possible bias. We  have added a human evaluation study to provide further analysis and validation of our metrics. Specifically,  we recruit 11 expert annotators, who are CS PhDs among 4 Universities’ CS departments and who are not authors of this paper. We collect over blind annotations and compute agreement scores between the human and LLM judges. Overall the results, demonstrate a high agreement rate of 83.33%. We have added these results to our manuscript (Section 6.3)
>
>
> > Q1. Reference Coverage Metric: How are “important” references determined in the human exemplar? Is this labeling done manually or via an automated heuristic/LLM? Clarifying this process (and whether alternative relevant sources are somehow acknowledged) would help us interpret the reference coverage scores.
>
> Thank you for raising this question, which is important to clarifying our retrieval quality metrics. Our Reference Coverage metric provides a notion of recall over a set of key, important sources. We carefully design this metric by curating a set of “important references” from each human exemplar using an LLM-judge which marks each extracted reference as “important” or “not important”. To validate this metric we have added a new human evaluation study, recruiting 11 expert annotators, who are CS PhD or postdocs among 4 Universities’ CS departments and who are not authors of this paper. We collect hundreds of blind annotations and report the confusion matrix of human-LLM agreement below for labeling references as “important”. Overall the results demonstrate a low likelihood of falsely penalizing systems, given by the low False Positive Rate (9.8%). Furthermore, we find a 65.9% agreement rate, demonstrating that our LLM-judge provides a strong heuristic for determining key references from the human exemplars without falsely penalizing systems that fail to cite non-essential references from the human exemplars. We also provide details of the specific prompt we use for our LLM-judge in the Appendix Figure 5. We have added these results to our manuscript (Section 6.3).
>
>
> | Actual / Predicted | Not Important  | Important  |
> |--------------------|--------|-------|
> | Not Important      | 40.2 % | 9.8%  |
> | Important         | 24.2%  | 25.7% |
>
> While our reference coverage metric does not reward alternative sources beyond the pre-determined “important” references, our other retrieval metrics _do_ allow for alternative references. Specifically, Retrieval Rate, uses an LLM-judge to assign graded-relevance scores to any retrieved source returned by the system-generated report and Document Importance rewards reports with highly cited papers using a normalized score that does not require sources to match our pre-determined “important” references.
>
> > W2. Reference coverage limitations: The metric penalizes systems for citing valid alternative papers not in the human reference list, rather than evaluating whether different sources cover the same content. Consider assessing content coverage instead of exact reference matching.
>
> Thank you for this point, which raises an important clarification regarding our evaluation metrics. We clarify that we evaluate _content coverage_ using our Nugget Coverage metric. As a result, Nugget Coverage and Reference Coverage provide complementary metrics, where Nugget Coverage assesses _factual coverage_ and Reference Coverage assess a notion of reference coverage or recall of “important” sources.

---

> ### Author Response · Authors · 2025-11-25
> **Response to Reviewer ijFd (2/3)**
>
> > W3. Narrow scope: The dataset covers only 63 CS queries from one quarter. Expanding to other domains (biomedical, social sciences) and creating a fixed benchmark subset alongside monthly updates would improve generalizability and enable longitudinal comparisons.
>
> Thanks you for this constructive feedback, and we agree that domain coverage is a crucial aspect of our benchmark. We have addressed this by adding an additional study over an expanded dataset, DeepScholar-Nov-2025. This dataset includes 200 queries scraped from a diverse set of over 75 arxiv domains, which include both papers from ArXiv’s CS, Biology, Physics, Economics and Finance servers. We report the performance of high-performing baselines with both open-source and closed-source models below and have added these results to our manuscript (Appendix 8.3.2):
>
> | Model                          | Org   | Nug   | Rel Rate | Ref Cov | Doc Imp | Cite P | Claim Cov |
> |--------------------------------|-------|-------|----------|---------|---------|--------|-----------|
> | DeepScholar (Llama-4-Scout)    | 0.120 | 0.358 | 0.395    | 0.072   | 0.082   | 0.178  | 0.581     |
> | DeepScholar (GPT4.1 + o3)      | 0.578 | 0.479 | 0.568    | 0.087   | 0.056   | 0.563  | 0.578     |
> | Search AI (Llama-4-Scout)      | 0.108 | 0.252 | 0.179    | 0.034   | 0.079   | 0.189  | 0.314     |
> | Search AI (o3)                 | 0.608 | 0.480 | 0.623    | 0.078   | 0.034   | 0.475  | 0.452     |
>
> We highlight that our work proposes an automated dataset-curation pipeline, rather than a fixed dataset, and additional domains can be easily configured by our pipeline for practitioners to run. Our core analysis and main results in the paper remain on the DeepScholar-June-2025, which focuses on computer science papers. This choice was deliberate in order to allow us to perform systematic validation of our benchmark metrics and design with expert annotators consisting of CS PhDs and postdocs.
>
> Notably, we observe patterns from our evaluation on DeepScholar-Nov-2025 that are consistent with our main results on DeepScholar-June-2025. For both DeepScholar-ref and the Search Agent baseline, the o3-based variants substantially outperform their Llama-4-Scout counterparts on Organization, Nugget Coverage, Relevance Rate, Citation Precision, and Claim Coverage, while Reference Coverage and Document Importance remain broadly low across baselines. These trends mirror the relative ranking and qualitative gaps seen in our main benchmark slice analyzed in the paper, suggesting that our conclusions from DeepScholar-June-2025 generalize beyond queries related to Computer Science.
>
> > Q2. Generality: The dataset currently draws from computer science papers. Do the authors plan to extend DeepScholarBench to other research domains (e.g. medicine, physics), or is the focus primarily on CS/AI literature? Expanding domain coverage could increase the benchmark’s impact.
>
> Yes, we plan to expand additional domains to future datasets, and have done so to create the DeepScholar-Nov-2025 dataset, described above, to demonstrate our benchmark’s generalizability. The main analysis in our paper focuses on the CS literature to allow us to perform rigorous validation and human evaluation studies, for which we recruit CS PhDs and postdocs to provide expert annotations.
>
> > Q3. Live Benchmark Dynamics: With monthly updates, how will the authors ensure fair comparison over time? For example, if a new model is evaluated on a future set of queries, will there be a way to compare its performance to the scores reported in this paper (on the June 2025 set)? It would be useful to know if a static test subset or leaderboard versioning is planned to handle the evolving data.
>
> Thank you for your question on this important component of our live benchmark. We plan to version our leaderboard, providing successive datasets, each with recent papers from the last few days to months. We believe this design is important to prevent data contamination since recent systems and models could be trained on papers from older dataset version.

---

> > ### Author Response · Authors · 2025-11-25
> > **Response to Reviewer ijFd (3/3)**
> >
> > > W4. Weak citation detection: The baseline (DeepScholar-ref) scores near 0% on Document Importance, missing high-impact papers. Systems should incorporate citation metrics or seminal paper detection to identify influential references.
> >
> > > Q4. Baseline Pipeline Improvements: DeepScholar-ref performs well, but notably struggles with retrieving highly-cited documents (Document Importance ≈0.01). What are the authors’ thoughts on augmenting the pipeline with an “importance awareness” (e.g. factoring in citation counts or influence of sources during retrieval)? This could potentially boost that metric – have such improvements been considered for future versions of the reference pipeline?
> >
> > We thank the reviewer for this feedback and question – we agree that low Document Importance scores is a key limitation of DeepScholar-ref as well. In this work, we focus our contribution on our automated data pipeline, metrics and live benchmark. Rather than explicitly optimizing our reference pipeline, this work instead aims to provide a relatively simple, strong, open-source baseline as a foundation for future work, noting opportunities for improvement. We agree that “importance awareness” is a very interesting and important direction for future work!

---

### Official Review · Reviewer_KS8o · 2025-11-01

**Soundness:** 3
**Presentation:** 2
**Contribution:** 3
**Rating:** 6
**Confidence:** 3

**Summary:**

This paper introduces DeepScholar-bench, a new benchmark designed to evaluate AI systems on the task of generative research synthesis. The core idea is to create a "live" benchmark that stays current by automatically sourcing new queries from recent, high-quality ArXiv papers. The specific task is to generate a "Related Work" section for a given paper's title and abstract. The authors propose a comprehensive, automated evaluation framework that assesses performance across three key dimensions: Knowledge Synthesis, Retrieval Quality, and Verifiability. They also contribute DeepScholar-ref, an open-source reference pipeline. Through a systematic evaluation of 14 existing systems, including open-source models, commercial search agents, and OAI's DeepResearch, the paper demonstrates that the task is far from solved, with no system surpassing a geometric mean of 31% across all metrics.

**Strengths:**

1. The paper addresses a critical and timely problem. s AI systems for long-form, knowledge-intensive tasks become more common and rigorous.

2. The evaluation framework is comprehensive with multiple fine-grained metrics (Table 1). The authors perform manual validation for their LLM-based metrics, showing high agreement (70-82%) with human judgments (Table 7, lines 963-968).

**Weaknesses:**

- Seems contradiction in the  scope of retrieval: The paper motivates the task by emphasizing retrieval from the "live web" (lines 015, 050). However, the experimental setup explicitly restricts all systems to retrieving only from the ArXiv API (line 361). The benchmark may not be fully evaluating the systems' ability to navigate the complexities of the true "live web."

- The presentation in Table 2 of human-written exemplars receiving very low scores for Citation Precision (.278) and Claim Coverage (.205) in a main results table next to AI systems that score as high as .944 (Cite-P) and .937 (Claim Cov) can be misleading, though a footnote (lines 345-348) was given that that this is a severe underestimate.

**Questions:**

1. Given the acknowledged issues with automatically measuring the verifiability of human writing, have you considered performing a small-scale manual evaluation on a subset of the human-written exemplars? I'd say that even with a small sample would provide a much clearer "gold standard" for the verifiability metrics.

2. The cost and latency analysis in Table 5 is excellent and highly valuable. However, costs are not reported for the proprietary Search Agent baselines (e.g., those using GPT-4.1, Claude, etc.). Could you add these figures to provide a complete and fair comparison?

---

> ### Author Response · Authors · 2025-11-25
> **Response to Reviewer KS8o (1/2)**
>
> > W1. Seems contradiction in the scope of retrieval: The paper motivates the task by emphasizing retrieval from the "live web" (lines 015, 050). However, the experimental setup explicitly restricts all systems to retrieving only from the ArXiv API (line 361). The benchmark may not be fully evaluating the systems' ability to navigate the complexities of the true "live web."
>
> Thank you for raising this question regarding the search API exposed to each baseline in our benchmark setting. We would like to clarify that we control the search API in order to provide a fair and reproducible benchmark. Our task still requires systems to navigate this large, continually evolving web corpus of “the live web”, albeit a subset of all web data available focused on scientific search. To compare the performance of systems limited to academic web search via the ArXiv API to systems with access to a general web search API, such as tavily.com or parallel.ai, we provide a comparison in Table 3. The results demonstrate that our benchmark setup using search via the ArXiv API provides a sufficiently difficult task, with performance results comparable to setups with general web search APIs. We are happy to revise the presentation in our manuscript further if the term "live web" may currently be unclear.
>
>
> Table 3
> |                                | **Knowledge Synthesis** |                 | **Retrieval Quality** |            |            | **Verifiability** |                  | **Geo. Mean** |
> |--------------------------------|--------------------------|-----------------|------------------------|------------|------------|--------------------|------------------|---------------|
> |                                | Org                      | Nug. Cov.       | Rel. Rate              | Ref Cov.   | Doc Imp.   | Cite-P            | Claim Cov (w=1)  |               |
> | **_DeepScholar-ref (GPT-4.1, Claude)_** |                          |                 |                        |            |            |                    |                  |               |
> | arxiv.org Retrieval            | 0.698                    | 0.307           | 0.610                  | 0.152      | 0.009      | 0.944              | 0.895            | 0.286         |
> | parallel.ai Retrieval          | 0.865                    | 0.444           | 0.675                  | 0.160      | 0.017      | 0.846              | 0.781            | 0.334         |
> | taviliy.com Retrieval          | 0.929                    | 0.327           | 0.550                  | 0.070      | 0.015      | 0.711              | 0.578            | 0.258         |
> | Oracle Retrieval (arxiv.org)   | 0.782                    | 0.487           | 0.686                  | 1.000      | 1.000      | 0.955              | 0.899            | 0.808         |
> | Oracle Retrieval (All)         | 0.778                    | 0.528           | 0.680                  | 1.000      | 0.822      | 0.941              | 0.828            | 0.782         |
> | **_DeepScholar-ref (Llama-4)_**          |                          |                 |                        |            |            |                    |                  |               |
> | arxiv.org Retrieval            | 0.206                    | 0.241           | 0.436                  | 0.103      | 0.008      | 0.674              | 0.851            | 0.195         |
> | parallel.ai Retrieval          | 0.246                    | 0.265           | 0.559                  | 0.114      | 0.015      | 0.223              | 0.543            | 0.186         |
> | taviliy.com Retrieval          | 0.111                    | 0.229           | 0.532                  | 0.030      | 0.016      | 0.442              | 0.676            | 0.153         |
> | Oracle Retrieval (arxiv.org)   | 0.202                    | 0.316           | 0.681                  | 1.000      | 1.000      | 0.658              | 0.868            | 0.590         |
> | Oracle Retrieval (All)         | 0.198                    | 0.350           | 0.693                  | 1.000      | 0.822      | 0.796              | 0.890            | 0.600         |

---

> ### Author Response · Authors · 2025-11-25
> **Response to Reviewer KS8o (2/2)**
>
> > W2. The presentation in Table 2 of human-written exemplars receiving very low scores for Citation Precision (.278) and Claim Coverage (.205) in a main results table next to AI systems that score as high as .944 (Cite-P) and .937 (Claim Cov) can be misleading, though a footnote (lines 345-348) was given that that this is a severe underestimate.
>
> > Q1. Given the acknowledged issues with automatically measuring the verifiability of human writing, have you considered performing a small-scale manual evaluation on a subset of the human-written exemplars? I'd say that even with a small sample would provide a much clearer "gold standard" for the verifiability metrics.
>
> Thank you for this suggestion, and we have performed an additional manual evaluation of human exemplars, and we compute scores of 90% Citation Precision and 85% Citation Coverage from a sample of 20 papers. We have updated the manuscript to include these numbers and aid the reader in interpreting our results in Table 2.
>
> > Q2. The cost and latency analysis in Table 5 is excellent and highly valuable. However, costs are not reported for the proprietary Search Agent baselines (e.g., those using GPT-4.1, Claude, etc.). Could you add these figures to provide a complete and fair comparison?
>
> Thank you for this question, and to clarify, we provide the Search Agent cost and latency numbers in the manuscript in Table 5, which are as follows:
>
> | Model                   | Latency (s) | Cost ($) |
> |-------------------------|-------------|----------|
> | Search Agent (GPT-4.1)  | 39          | 0.07     |
> | Search Agent (o3)       | 263         | 0.15     |
> | Search Agent (Claude)   | 147         | 1.36     |
> | Search Agent (Gemini)   | 442         | 0.11     |

---

### Official Review · Reviewer_9gFY · 2025-11-04

**Soundness:** 2
**Presentation:** 3
**Contribution:** 2
**Rating:** 4
**Confidence:** 4

**Summary:**

The paper introduces DeepScholar-Bench,a benchmark and automated evaluation framework for generative research synthesis.Evaluates systems along three axes — synthesis, retrieval, verifiability — using seven automated metrics. The authors also release DeepScholar-Ref, an open-source reference pipeline built on LOTUS.

**Strengths:**

- The benchmark design, data pipeline, and multi-axis metric structure are coherent.
- Clear task decomposition: synthesis / retrieval / verifiability.
- Open-source infrastructure and a reproducible reference pipeline DeepScholar-Ref.
- Timely motivation — evaluation gap in long-form research reasoning
- Clean writing, organized figures and tables
- Logical experiment and ablations (oracle, retriever variants) with consistent metric and reporting

**Weaknesses:**

- No quantitative human–metric correlation (claimed “manual validation” only)
- Heavy reliance on LLM-as-judge introduces potential bias, no transparency or calibration details, it’s really unclear whether the metrics reflect the real quality. No human/user evaluation to confirm synthesis usefulness
- Dataset scale is small (63 ArXiv CS papers) and domain coverage is narrow, which limits the robustness and generalizability of the benchmark
- The novelty is incremental vs. prior deep-research benchmarks
- Tables are kinds of dense, with limited analysis, no significance reproted

**Questions:**

- How stable are results across LLM-judge versions or calibration settings? What's the correlations between automated metrics and human ratings
- How are “important references” defined for Reference Coverage?
- How to prevent that the models have been trained on the same ArXiv data used for benchmark construction?

---

> ### Author Response · Authors · 2025-11-25
> **Response to Reviewer 9gFY (1/4)**
>
> > W1. No quantitative human–metric correlation (claimed “manual validation” only)
>
> > W2. Heavy reliance on LLM-as-judge introduces potential bias, no transparency or calibration details, it’s really unclear whether the metrics reflect the real quality. No human/user evaluation to confirm synthesis usefulness
>
> > Q1. How stable are results across LLM-judge versions or calibration settings? What's the correlations between automated metrics and human ratings
>
> Thank you for raising these important concerns regarding our use of LLM-judges. We have added two experimental studies to address these points: (a) we conduct a blind human evaluation study, recording the human-LLM agreement, and (b) we evaluate different LLM-judges providing analysis on the effect of different model judges. We provide details of each below and have added these studies to our manuscript (Section 6.3). Additionally, we provide all prompts used for our LLM judges in Appendix Section 8.4, and we implement each without calibration.
>
> __Human Evaluation Study__
> To validate the overall quality of our LLM-based metrics, we recruit 11 expert annotators, who are CS PhD among 4 Universities’ CS departments and who are not authors of this paper. We collect over 300 blind annotations for the three key LLM-based evaluation tasks introduced by our work: labeling nuggets as  “vital” or “not vital” for writing related work section task, labeling references as “important” or “not important” for writing a related work section, and pairwise comparisons to evaluate organization of a related works section. Our results, shown below, demonstrate high agreement rates that validate the quality of our automated metrics. We also note that our LLM-judge for Reference Coverage exhibits a low false positive rate of 9.8% (i.e., where the LLM labels a non-important reference as important), indicating a low-likelihood of falsely penalizing systems.
> | **Evaluation Metric**        | **Labels**                                   | **Agreement Rate Between Human / LLM** |
> |------------------------------|-----------------------------------------------|---------------------|
> | Nugget Importance            | Nugget Label (Vital / Okay / Irrelevant)                    | 71.43%                    |
> | Reference Coverage           | Reference Importance (Not Imp. / Imp.)       | 65.90%              |
> | Organization                 | Pairwise Comparison (Lose / Tie / Win)       | 83.33%              |
>
>
> We note that our evaluation additionally leverages other LLM-based evaluation tasks for our Relevance Rate, Claim Coverage, Citation Precision and Nugget Coverage metrics, which build on existing work that shows LLM judges agree well with humans on those metrics. For these metrics, high human-LLM agreement rates have been validated by prior work that studies LLM-judges for assigning graded relevance scores (Faggioli et al., 2023; Rahmani et al., 2024b; Thomas et al., 2024; Asai et al., 2024), assessing entailment relations (Gao et al., 2023; Liu et al., 2023) and assessing nugget coverage (Pradeep et al., 2025; Upadhyay et al., 2024b;a; Faggioli et al., 2023; Rahmani et al., 2024a). We additionally provide a manual validation of these metrics, reported in the Appendix Table 7, demonstrating strong agreement rates, above 70%. Our manual validation consists of 400 human annotations from our pilot study of 3 CS PhD students or postdocs.

---

> ### Author Response · Authors · 2025-11-25
> **Response to Reviewer 9gFY (2/4)**
>
> __LLM Judge Agreement Study__
> We evaluate correlation between different LLM-judges by comparing our chosen GPT-4o judge with alternative open-source judges, Llama-4-17b-16e-Instruct and DeepSeek-R1-Distill-Qwen-32B. We evaluate all metrics using each judge over a subset of 5 high-performing open-source and closed source baselines. We report Pearson correlation between different LLM-as-a-judge evaluations of the baselines across metrics. We also run a permutation test and report the statistical significance of the resulting correlations. We note that Document Importance is excluded from the table below since it does not use an LLM-judge. Overall, the results demonstrate a relatively strong positive correlation between different model pairs across each respective metric. We have added this study to our manuscript  in  section 8.3.6 Agreement between different LLM Judges.
>
>
> Table 11. Pearson correlations between system-level scores produced by different LLM judges across five baselines for each metric.
>
> | Metric                          | Org.  | Nug. Cov. | Rel. Rate | Ref. Cov. | Cite-P | Claim Cov (w=1) |
> |---------------------------------|-------|-----|-----------------|---------|--------|------------------|
> | **Pearson Corr (Gpt-4o, Llama-4)**        | 0.985 | 0.413 | 0.941 | 0.975 | 0.817 | 0.958 |
> | **P-value (Gpt-4o, Llama4)**             | 0.025 | 0.256 | 0.066 | 0.017 | 0.041 | 0.017 |
> | **Pearson Corr (Gpt-4o, Deepseek-r1)**   | 0.966 | 0.920 | 0.980 | 0.990 | 0.843 | 0.996 |
> | **P-value (Gpt-4o, Deepseek-r1)**        | 0.041 | 0.041 | 0.017 | 0.017 | 0.033 | 0.017 |
> | **Pearson Corr (Llama-4, Deepseek-r1)**  | 0.991 | 0.668 | 0.942 | 0.994 | 0.986 | 0.963 |
> | **P-value (Llama-4, Deepseek-r1)**       | 0.025 | 0.124 | 0.041 | 0.017 | 0.058 | 0.017 |

---

> ### Author Response · Authors · 2025-11-25
> **Response to Reviewer 9gFY (3/4)**
>
> > W3.  Dataset scale is small (63 ArXiv CS papers) and domain coverage is narrow, which limits the robustness and generalizability of the benchmark
>
> Thank you for this constructive feedback, and we agree that domain coverage is a crucial aspect of our benchmark. We have addressed this by adding an additional study over an expanded dataset, DeepScholar-Nov-2025, demonstrating the generalizability of our benchmark. This dataset includes 200 queries scraped from a diverse set of over 75 arxiv domains, which include both papers from ArXiv’s CS, Biology, Physics, Economics and Finance servers. We report the performance of high-performing baselines with both open-source and closed-source models below and have added these results to our manuscript (Appendix 8.3.2):
>
> Table 8. Performance of Selected Systems on DeepScholar-Nov-20255 (200 queries across >75 arXiv subject areas)
> | Model                          | Org   | Nug   | Rel Rate | Ref Cov | Doc Imp | Cite P | Claim Cov |
> |--------------------------------|-------|-------|----------|---------|---------|--------|-----------|
> | DeepScholar (Llama-4-Scout)    | 0.120 | 0.358 | 0.395    | 0.072   | 0.082   | 0.178  | 0.581     |
> | DeepScholar (GPT4.1 + o3)      | 0.578 | 0.479 | 0.568    | 0.087   | 0.056   | 0.563  | 0.578     |
> | Search AI (Llama-4-Scout)      | 0.108 | 0.252 | 0.179    | 0.034   | 0.079   | 0.189  | 0.314     |
> | Search AI (o3)                 | 0.608 | 0.480 | 0.623    | 0.078   | 0.034   | 0.475  | 0.452     |
>
>
> We highlight that our work proposes an automated dataset-curation pipeline, rather than a fixed dataset, and additional domains can be easily configured by our pipeline for practitioners to run. Our core analysis and main results in the paper remain on the DeepScholar-June-2025, which focuses on computer science papers. This choice was deliberate in order to allow us to perform systematic validation of our benchmark metrics and design with expert annotators consisting of CS PhDs and postdocs.
>
> Notably, we observe patterns from our evaluation on DeepScholar-Nov-2025 that are consistent with our main results on DeepScholar-June-2025. For both DeepScholar-ref and the Search Agent baseline, the o3-based variants substantially outperform their Llama-4-Scout counterparts on Organization, Nugget Coverage, Relevance Rate, Citation Precision, and Claim Coverage, while Reference Coverage and Document Importance remain broadly low across baselines. These trends mirror the relative ranking and qualitative gaps seen in our main benchmark slice analyzed in the paper, suggesting that our conclusions from DeepScholar-June-2025 generalize beyond queries related to Computer Science.
>
> > W4. The novelty is incremental vs. prior deep-research benchmarks
>
> We highlight that our work, in contrast to prior work, is the only one to offer a live benchmark for long-form research synthesis. Prior deep-research benchmarks fall into one of two categories: either they use expert-curated datasets or provide short-form question-answering benchmarks. Prior works proposing expert-curated benchmarks quickly become outdated, risk data-contamination, and are difficult to scale. These include QABench (Asai et al., 2024), OpenResearcher (Zheng et al., 2024), DeepConsult (you.com, 2025), ResearcherBench (Xu et al., 2025), DeepResearch Bench (Du et al., 2025), Deep Research Bench (FutureSearch et al., 2025), SurGE (Su et al., 2025), and LiveDRBench (Java et al., 2025). Our work avoids this pitfall by designing an automated dataset curation pipeline for a real research synthesis task: generating related works sections. On the other hand, existing short-form factuality benchmarks, such as SimpleQA (Wei et al., 2024), FRAMES (Krishna et al., 2025), GAIA (Mialon et al., 2023), BrowserComp (Wei et al., 2025), BrowserComp-Plus (Chen et al., 2025) WebWalkerQA (Wu et al., 2025), DeepResearch Arena (Wan et al., 2025) are limited to short, easily verifiable answers and do not address the challenges of evaluating long-form research reports, which our work addresses through a holistic, automated evaluation framework, which we validate with human evaluation.
>
> > W5. Tables are kinds of dense, with limited analysis, no significance reproted
>
> Thank you for this feedback – we have improved our results section and its presentation, updating our manuscript to report the significance of the best-performing baseline for each metric using a two-sided t-test in Table 2 and Table 3.

---

> ### Author Response · Authors · 2025-11-25
> **Response to Reviewer 9gFY (4/4)**
>
> > Q2. How are “important references” defined for Reference Coverage?
>
> Thank you for your question. We clarify that we extract all cited references from the human exemplars and use an LLM-judge to mark each one as “important” or “not important”. To provide further validation of our LLM-judge labels for this metric, we have added a human evaluation study. Specifically, we recruit 11 expert annotators, who are CS PhDs across 4 Universities’ CS departments and who are not authors of this paper. We collect over 100 blind annotations for this metric and report the confusion matrix of human-LLM agreement below. Overall the results demonstrate that our LLM-judge provides a strong heuristic for determining key references from the human exemplars, with a 65.9% agreement rate. Furthermore, the confusion matrix shows our metric has a low likelihood of falsely penalizing systems, given by the low False Positive Rate (9.8%). We have updated our manuscript with details of our human study (Section 6.3), and have provided the specific prompt we use for our LLM-judge in the Appendix Figure 5.
>
> Table 4:
> | Actual / Predicted | Not Important  | Important  |
> |--------------------|--------|-------|
> | Not Important      | 40.2 % | 9.8%  |
> | Important         | 24.2%  | 25.7% |
>
> > Q3. How to prevent that the models have been trained on the same ArXiv data used for benchmark construction?
>
> Thank you for this question raising an important point about our benchmark design to avoid data contamination. Our automated data pipeline allows us to easily construct new datasets and configure paper publication dates to scrape from. To provide a live, contamination free benchmark, each month we plan to create a new dataset instantiation configured with papers from the past few days to weeks. We will configure this date range to be _after_ the release date of the models in our benchmarked systems to prevent training data contamination.

---

> > ### Comment · Reviewer_9gFY · 2025-11-26
> >
> > Thanks for the response. Adding the human evaluation is a major revision needed. I’d give higher rating given more details of human evaluation. I leaning towards resubmitting to next cycle.

---

> ### Author Response · Authors · 2025-12-03
> **Response to Reviewer 9gFY**
>
> Thank you for the feedback and we have conducted the requested human evaluation (described above). We have updated our manuscript to include further details of our systematic study in Sections 6.3 and Appendix 8.3.4. Specifically, we conduct a human evaluation with 11 annotators, all Computer Science PhD students from four research universities across North America.  In total, we collect over 300 human annotations from a blind study conducted via a Qualtrics Survey. The results validate the LLM-based judges introduced by our automated evaluation for assessing knowledge synthesis and retrieval quality. Specifically, we observe a 71.43% agreement score for pairwise comparisons judging Organization, a 83.33% agreement score for nugget labeling to compute Nugget Coverage, and a 65.9% agreement score for labeling reference importance to compute Reference Coverage. We describe our setup in detail below and provide detailed confusion matrices displaying the results of our study in Table 4.
>
> **Organization**
> For evaluating our Organization metric, we sample generated reports and show annotators the human-written related work section alongside a system-generated report. For each pair, annotators indicate whether they prefer a system-generated report, prefer the human-written reference, or consider them a tie. All annotations are blind. These labels are used to evaluate the pairwise comparison outcomes underlying our Organization metric. We compute agreement between human and LLM labels for each pairwise comparison.
>
> Agreement Results (Table 4):
> | **Human / LLM** | Reference | Generated | Tie    |
> | --------------- | --------- | --------- | ------ |
> | **Reference**   | 14.29%    | 0%        | 14.29% |
> | **Generated**   | 0%        | 57.14%    | 14.29% |
> | **Tie**         | 0%        | 0%        | 0%     |
>
>
> **Nugget Labeling**
> Our Nugget Coverage metric relies on an LLM to label generated as nuggets as "okay" or "vital. To study the soundness of this automated labeling, we first generate information nuggets from human-written related work sections. Annotators are then shown individual nuggets and asked to judge whether each nugget is 'vital' for understanding the paper, 'okay', or 'irrelevant'. These nugget-importance labels determine which nuggets are treated as essential when computing nugget coverage scores. We compute agreement between human and LLM labels for each nugget.
>
> Agreement Results (Table 4):
> | **Human / LLM** | Vital  | Okay   |
> | --------------- | ------ | ------ |
> | **Vital**       | 58.33% | 8.33%  |
> | **Okay**        | 8.33%  | 25.00% |
> | **Irrelevant**  | 0%     | 0%     |
>
>
>
> **Important Reference Labeling**
> Our Reference Coverage metric relies on an LLM to label which references from a human written exemplar section are considered "important" or a sound related works section. In our study, we ask each annotator to select a high quality paper whose related work section they are comfortable with. For that paper, the annotator identifies at least six references they consider important (i.e., references that should appear in a good related work section) and at least six references they consider not important (i.e., references that could be omitted or substituted without harming the quality of the section). These labels form gold sets of important versus non-important references, which we use both to evaluate Reference Coverage and to validate our LLM-based importance labels. We compute agreement between human and LLM labels for these references.
>
> Agreement Results (Table 4):
> | **Human / LLM** | Not Imp. | Imp.  |
> | --------------- | -------- | ----- |
> | **Not Imp.**    | 40.2%    | 9.8%  |
> | **Imp.**        | 24.2%    | 25.7% |
>
> We note that our evaluation additionally leverages other LLM-based evaluation tasks for our Relevance Rate, Claim Coverage, Citation Precision and Nugget Coverage metrics, which build on existing work that shows LLM judges agree well with humans on those metrics. For these metrics, high human-LLM agreement rates have been validated by prior work that studies LLM-judges for assigning graded relevance scores (Faggioli et al., 2023; Rahmani et al., 2024b; Thomas et al., 2024; Asai et al., 2024), assessing entailment relations (Gao et al., 2023; Liu et al., 2023) and assessing nugget coverage (Pradeep et al., 2025; Upadhyay et al., 2024b;a; Faggioli et al., 2023; Rahmani et al., 2024a). We additionally provide a manual validation of these metrics, reported in the Appendix Table 7, demonstrating strong agreement rates, above 70%. Our manual validation consists of 400 human annotations from our pilot study of 3 CS PhD students or postdocs.

---

### Author Response · Authors · 2025-11-25
**Response Summary**

We extend our sincere gratitude to the reviewers for their thoughtful evaluation of our submission and their constructive, insightful feedback. We are particularly encouraged by their recognition of our work’s **strong motivation** (@9gFY, @KS8o, @ijFd, @s3EJ, @AnFP), **novel benchmark design** (@ijFd  @AnFP @ozAV), **well-justified, comprehensive evaluation framework** (@9gFY @KS8o @ijFd @s3EJ @ozAV), **careful, high-quality methodology** (@ijFd, @s3EJ), **thorough empirical analysis** (@9gFY @s3EJ) and **clear presentation** (@9gFY @ijFd @s3EJ @ozAV).

We have carefully read each reviewer's comments, and in response to the insightful feedback, we have made substantial revisions to strengthen the manuscript. We conducted a series of new experiments and analysis and summarize major updates as follows:
- **Human Evaluation**: We have conducted a new human evaluation study, validating the effectiveness of the LLM-judges used in our evaluation, particularly for assessing Reference Coverage, Organization, and Nugget Coverage metrics (@9gFY, @ijFd, @AnFp)
- **New, Expanded Dataset**: We have created a new, expanded dataset, DeepScholar-Nov-2025 containing 200 papers from a diverse set of over 75 domains across from ArXiv’s CS, Biology, Physics, Economics and Finance servers, demonstrating the generalizability of our dataset pipeline, benchmark, and conclusions. (@9gFY, @ijFd, @s3EJ)
- **Additional Ablation on Query Sensitivity**: We have conducted a new ablation study providing a controlled experiment to test different types of query inputs, e.g., a paper abstract, two-sentence description, or research query. Overall, we find high correlation of performance results among each of these query variations, solidifying the generalizability of our benchmark task using a paper’s abstract in each query. (@AnFp)
- **Statistical Analysis**: We perform a two-sided t-test, demonstrating the statistical significance of the best-reported baseline in our main results for each metric. (@9gFY, @s3EJ)
- **Manual Validation of Verifiability for Human Exemplars**: We perform an additional manual validation, computing the citation precision and claim coverage of human-written exemplar related works sections. We add this estimate to our main results to improve the presentation and interpretability of our verifiability scores. (@KS8o, @s3EJ)


We provide individual responses to each reviewer below and have incorporated all discussed updates to our manuscript with blue text.

---

### Meta-Review · Area_Chair_ZGuZ · 2026-01-08

**Summary:**

While the motivation to create a "live" benchmark is appreciated, the primary concern driving this decision is the limited scale and scope of the dataset. As highlighted by Reviewer 9gFY and Reviewer s3EJ, the initial dataset consists of only 63 ArXiv papers. Even with the supplementary expansion, the scale remains insufficient to serve as a robust, meaningful foundation for the general research community. For a benchmark paper to clear the acceptance bar, the data foundation must be statistically significant and broad enough to support rigorous evaluation, which this submission currently lacks. We encourage the authors to significantly scale up the data collection and broaden the source diversity beyond ArXiv for future iterations. Therefore, I recommend to reject this paper and encourage the authors to resubmit to a future venue.

**Reviewer Concerns:**

Addressed:
-  Query Sensitivity: The authors addressed Reviewer AnFp’s concern regarding the realism of using full abstracts as queries by conducting an ablation study with alternative query formats (e.g., two-sentence summaries).
- Human Validation: The lack of human validation raised by Reviewer 9gFY and Reviewer ijFd was mitigated by a new human evaluation study showing reasonable agreement between LLM judges and human annotators.

 Outstanding:
- Dataset Scale and Quality: Reviewer 9gFY and Reviewer s3EJ raised critical concerns about the small sample size (63 papers) and narrow domain coverage (CS papers only). Despite the addition of a 200-paper set in the rebuttal, the core analysis relies on a dataset that is viewed as too small to ensure statistical robustness or meaningful generalization for the wider community.
- Retrieval Scope: Reviewer KS8o noted a contradiction between the "live web" motivation and the experimental restriction to the ArXiv API. This limits the paper’s claim of evaluating systems on the complexity of the true live web.
 - Metric Bias: Reviewer s3EJ maintained concerns regarding field and age bias in the "Document Importance" metric, which may disadvantage emerging domains regardless of normalization attempts.

**Reviewer Scores:**

Some reviewers are willing to maintain the rejection score. In general, despite the acceptance recommendation, there are a substantial portion of reviewers are negative.

---

### Decision · Program_Chairs · 2026-01-26

Reject